# Organic carbon rich sediments: benthic foraminifera as bio-indicators of depositional environments

Elena Lo Giudice Cappelli[1], Jessica L. Clarke[1,2], Craig Smeaton[1], Keith Davidson[3], William E. N. Austin[1,3]

[1]School of Geography and Sustainable Development, University of St Andrews, St Andrews, KY16 9AL, Scotland, UK

[2]School of Natural and Environmental Sciences, Newcastle University, Newcastle, NE1 7RU, UK

[3]Scottish Association for Marine Science, Scottish Marine Institute, Oban, PA37 1QA, Scotland, UK

*Correspondence to*: Elena Lo Giudice Cappelli (elena.logiudice.cappelli@gmail.com)

**Abstract.** Fjords have been described as hotspots for carbon burial, potentially playing a key role within the carbon cycle as climate regulators over multiple timescales. Nevertheless, little is known about the long-term fate of the carbon that may become stored in fjordic sediments. One of the main reasons for this knowledge gap is that carbon arriving on the seafloor is prone to post-depositional degradation, posing a great challenge when trying to discriminate between an actual change in carbon deposition rate and post depositional carbon loss. In this study, we evaluate the use of modern benthic foraminifera as bio-indicators of organic carbon content in six voes (fjords) on the west coast of Shetland. Benthic foraminifera are known to be sensitive to changes in organic carbon content in the sediments, and changes in their assemblage composition therefore reflect synchronous variations in the quantity and quality of carbon reaching the seafloor. We identified four environments based on the relationship between benthic foraminiferal assemblages and organic carbon content in the sediments: 1) Land-locked regions influenced by riverine/freshwater inputs of organic matter, namely the head of fjords with a restricted geomorphology; 2) Stressed environments with a heavily stratified water column and sediments rich in organic matter of low nutritional value; 3) Depositional environments with moderate organic content and mild or episodic current activity; 4) Marginal to coastal settings with low organic content, such as fjords with an unrestricted geomorphology. We conclude that foraminifera potentially provide a tool to disentangle primary organic carbon signals from post-depositional degradation and loss of organic carbon because of their environmental sensitivity and high preservation potential in the sedimentary record.

## 1 Introduction

Globally, fjords sequester about 18 Mt of organic carbon (OC) annually in their sediments, partly acting to reduce the remineralisation of OC into the environment and thus buffering the release of carbon dioxide ($CO_2$) to the atmosphere (e.g. Smith et al. 2015; Cui et al., 2016). A recent study by Smeaton et al. (2017) estimated that the sediments of Scotland's 111

major fjords hold 295 ± 52 Mt OC. In Shetland, these fjordic systems are different from their mainland counterparts, notably in that they are shallower and less glaciated (Edwards and Sharples, 1986) and peat or peaty soils dominate their catchments (Soil Survey of Scotland, 1981). Erosion of terrestrial organic matter (OM) is known to be a major source of OC in fjordic sediments (Cui et al., 2017) and Scottish peatlands hold 1620 Mt of OC (Chapman et al., 2009). Shetland peatlands cover over one third of the archipelago's total area and their erosion accounts for a loss of 1 to 4 cm yr$^{-1}$ from peat surfaces (Birnie, 1993). Eroding peatlands may therefore hold the potential to contribute significantly to OC burial and accumulation in Shetland's voes and to the overall coastal OC store of this region. Nevertheless, little is known about the long-term fate of the OC that may accumulate and become stored in these sediments. One of the main reasons for this knowledge gap is that OC is prone to post-depositional modification and degradation. It is therefore difficult to discern whether a down-core decrease in OC content is due to an actual change in carbon deposition over time or simply to this post depositional loss (Hülse et al., 2017; Stolpovsky et al., 2017).

Benthic foraminiferal assemblages have long been used as indicators of environmental change and to reconstruct the intrinsic characteristics of marine ecosystems. A recent study (Duffield et al., 2017) explored the relationship between foraminiferal assemblages and variations in OC content in the surficial and recent sediments of Norwegian fjords, highlighting temporal trends in OC fluxes and ecological quality status. In this study, we evaluate the use of modern benthic foraminifera as bio-indicators of OC content in six voes (fjords) on the west coast of Shetland whose sedimentation dynamics are likely influenced by the erosion of OC rich peat from the surrounding catchments (Fig. 1). As benthic foraminifera are sensitive to changes in OC content in the sediments (Alve et al., 2016 and references therein), changes in their assemblage composition should reflect synchronous variations in the amount of OC reaching the seafloor. Given the high preservation potential of their tests in marine sediments (e.g. Murray, 2003; Thomas et al., 1995), foraminifera potentially offer a means to discriminate primary OC signals from post-depositional OC degradation and may provide a tool to validate studies and models reconstructing past changes in the carbon cycle (Hülse et al., 2017; Stolpovsky et al., 2017). Nevertheless, the relationship between OC and foraminiferal assemblages is not straightforward and warrants further investigation.

Early studies developed conceptual models explaining benthic foraminiferal microhabitat preferences in deep sea settings based on changes in the organic flux reaching the seafloor and bottom water oxygenation (Jorissen et al., 1995; Van Der Zwaan et al., 1999). These authors found that in well-oxygenated bottom waters, food is usually the limiting factor controlling the distribution of benthic foraminifera, whereas under eutrophic conditions, bottom water oxygenation may become the dominant control on the composition of foraminiferal assemblages (Jorissen et al., 1995; Van Der Zwaan et al., 1999). Typically, food availability is strongly connected to OC fluxes and the transport of organic matter to the seafloor. In this context, it becomes crucial to understand not only food availability in terms of the quantity of OM reaching the seafloor, but also to understand the quality of this food source. For example, refractory organic matter (ROM), often derived from reworked terrestrial sources, is known to be of low nutritional value for benthic foraminifera compared to the more bio-available form of labile organic matter (LOM) (e.g. Fontanier et al., 2008; Jorissen et al., 1998). It should also be noted that OC accumulation rate and bottom water oxygen concentrations are often related, with low oxygen concentrations reducing the reworking/remineralisation of

OM and thus favouring its preservation in marine sediments. In certain environments, this implies that similar OC fluxes could result in different OC accumulation rates depending on bottom water oxygenation, the latter defining the composition of benthic foraminiferal assemblages (Van Der Zwaan et al., 1999).

To investigate the relationships between sedimentary OC in six west Shetland voes and the associated changes in benthic foraminiferal assemblages, this study combines geochemical and micropalaeontological techniques to: 1) Fingerprint the source (terrestrial vs. marine) and quality (refractory vs. labile) of organic matter and the form (organic vs. inorganic) of sedimentary carbon. 2) Establish benthic foraminiferal distribution in Shetland's voes from recent surficial sediments. 3) Investigate the use of benthic foraminifera as bio-indicators of OC content in coastal sediments and their potential for palaeo-OC reconstruction purposes.

## 2 Material and Methods

### 2.1 Area of study and sampling strategy

Shetland is a subarctic archipelago covering a total area of 1466 km$^2$, lying about 170 km north of mainland Scotland, UK (Fig. 1). Several voes dissect Shetland's coastline, providing an extended route for the transport of OC rich material from land to sea (Cui et al., 2016). The current study focuses on six voes on Shetland's west coast: Clift Sound, Sand Sound, Olna Firth, Aith Voe, Busta Voe and Vaila Sound (Fig. 1, Table 1). These six voes are characterised by different geomorphologies, freshwater inputs, proximity to the open sea and intensity of local currents; together these characteristics determine the amount of carbon deposited in the sediments. Clift Sound lies south of Shetland, the islands of Trondra and East Burra to the west, has steep-sides and a relatively unrestricted geomorphology (Fig. 1). Two main types of soil are present in the catchment of Clift Sound: humus-iron podzols cover the islands of East Burra and Trondra, while peaty soils dominate the eastern coast of Clift Sound (Supplementary Fig. 1). Winds are generally from the southwest, can be channelled and result in intensified currents due to the geometry of the sound and the surrounding highlands. Tidal energy can also be channelled into more vigorous currents (Cefas, 2007a).

Sand Sound is somehow T-shaped, located in the southwest of Shetland and comprises three areas: the head of the voe, the inner basin and the outer basin (Fig. 1). The head of the voe, almost perpendicular to the inner and outer basins, is a sheltered area that receives freshwater from a significant number of rivers draining the surrounding land. Three main types of soil characterise this area: peaty gleys, peaty podzols and peaty rankers (Supplementary Fig. 1). Here, the voe is very shallow and drying can occur in its eastern and western periphery (Cefas, 2007b, 2008b). In contrast, the rest of the voe is much deeper. The inner basin gently slopes away reaching more than 20 m at its maximum depth; drying can still occur where this basin meets the head of the voe (Cefas, 2007b). The outer basin has steep sides, is 42 m deep at its deepest point and completely open to sea. A shallow sill divides the inner from the outer basins (Cefas, 2007b)

Busta Voe, Olna Firth and Aith Voe are part of a major inlet on the southern coastline of St Magnus Bay on the west coast of Shetland (Fig. 1). Olna Firth represents the eastern branch of the inlet and is roughly oriented east – west, it is the furthest away from the open sea (Fig. 1). A large area of this voe exceed 30 m water depth; the northern coastline gradually slopes into the voe, whereas the south side shows a steeper gradient. Olna Firth is classified as micro-tidal and due to the generally low
level of energy in this system, stratification may occur, especially during summer time (Cefas, 2013).

Aith Voe is the southern component of the inlet, with a north – south orientation (Fig. 1). Due to the hilly landscape (up to 100 m) surrounding the voe, Aith Voe is exposed to winds from the north, which can significantly alter local circulation and prevent stratification of the water column. Two types of soil surround Aith voe: peaty and organic soils (Supplementary Fig. 1); few streams drain these soils, with the largest of them discharging on the east coast of Aith Voe (Cefas, 2010).

Busta Voe lies in the northern part of the inlet and is oriented north – south, sheltered and with a maximum water depth of 39 m (Fig. 1). Three main types of soil surround this area: peaty gleys, peaty podzols and peaty rankers (Supplementary Fig. 1); two rivers drain a relatively small catchment area into Busta Voe. Tidal flow is weak in the voe and wind generated currents are predominant; stratification of the water column especially during warm periods can occur (Cefas, 2008a).

Vaila Sound lies on the western coast of Shetland, with the island of Vaila to the southwest and the isle of Linga in the middle
(Fig. 1). Three types of soil dominate the catchment of this voe: peaty gleys, peaty podzols and peaty rankers (Supplementary Fig. 1). Most streams discharge in the northern part of the sound; hence, locations in this area might become more affected by terrigenous inputs. Additionally, Linga offers protection from wind and currents, which could facilitate localised accumulation of terrigenous material. Shelter from strong winds is also provided by the island of Vaila to the south. Here the tidal range is small and the associated energy weak; however, Vaila Sound is connected to the Atlantic Ocean west and east of the island of
Vaila (Cefas, 2009).

The MV Moder Dy 2015 cruise in west Shetland surveyed the six voes in August 2015 (Table 1). Marine surface sediments were sampled using a Duncan and Associates Van Veen grab with a sampling area of 0.1 m$^2$. Twenty-three surface sediment samples were obtained by scraping the top layer (~ 1 cm thick) of each grab with a domestic spoon, samples were then stored in a cold-box in sealed plastic bottles.

An earlier field survey of Shetland voes carried out in August 2009 measured bottom water temperature (BWT), salinity (BWS) and oxygen (O$_2$) at the same locations as this study (Fig. 2); however, no data were collected in Vaila Sound. Oceanographic measurements were made using a SeaBird SBE 19 plus conductivity-temperature-depth (CTD) profiler with a dissolved oxygen probe. Bottom water temperature varies between 14.14 and 12.20 °C being on average 12.74 °C (Fig. 2a). Bottom water salinity remains almost constant across these voes ranging between 35 and 35.2 p.s.u. with the exception of MD15-06
where BWS reached a minimum of 34.7 p.s.u (Fig. 2b). Similarly, bottom waters are mostly well-oxygenated in all the voes (O$_2$ > 5mg l$^{-1}$) with the exception of Olna Firth where lower O$_2$ were observed (Fig. 2c).

## 2.2 Particle size measurement

Particle size analyses were performed on a sub-set of surface sediments using a Coulter LS230 particle size analyser to quantify the volume (%) of the various grain sizes in each sample. We identified three categories based on the International Organisation for Standardisation (ISO) scale: 'Clay' includes particles < 2 μm, 'Silt' particles between 2-63 μm, and 'Sand' particles > 63 μm (upper limit of 2000μm). Before analysis, sediment samples were digested using 30 % hydrogen peroxide to remove organic matter, and 10 % hydrochloric acid to remove carbonate, following a similar methodology to that outlined by Austin and Evans (2010). Digestions were carried out to obtain an insoluble residue useful for comparing sediments having very different carbonate and organic matter content, which otherwise might skew the measurements because of *in situ* processes.

## 2.3 Loss on ignition (LOI) analysis

Loss on ignition measurements were carried out to quantify the percentage of total organic matter (TOM), and the respective amounts of labile (LOM) and refractory organic matter (ROM). The initial combustion temperature was set to 250˚ C, as significant mass loss of LOM has been recorded at this temperature (Mook and Hoskin, 1982). Sediments were successively heated to 550˚ C to quantify ROM, as refractory terrestrial and aquatic OM will likely burn off at this temperature (Kristensen, 1990). About 1 g of dried sediment was precisely weighted into crucibles of known weight for each surface sample ($M_0$). These were ashed in a muffle furnace at 250 ˚C for 4 hours, cooled and weighted ($M_{250}$). Samples were then returned to the furnace at 550 ˚C for 4 hours, cooled and re-weighted ($M_{550}$). TOM, LOM and ROM were calculated as follow:

$$LOM = [(M_0-M_{250})/M_S]*100$$

$$ROM = [(M_{250}-M_{550})/M_S]*100$$

$$TOM = LOM + ROM$$

Where $M_s$ is the sample weight minus the crucible weight. Replicate samples were measure at stations MD15-01A and 01B and MD15-05A and 05B which resulted in a mean relative error of ± 0.07 % for LOM, ± 0.06 % for ROM and ± 0.03 % for TOM, pointing to very good data reproducibility and representation of local conditions.

## 2.4 Bulk carbon elemental & stable isotope analyses

Surface sediment samples were analysed to determine bulk elemental C and stable isotope ($\delta^{13}C$) values. Each sediment sample was freeze dried and homogenized and approximately 12 mg of milled sediment were placed into silver capsules and a further 10 mg into tin capsules.

The samples encapsulated in silver underwent an acid fumigation step (Harris et al., 2001) to remove inorganic carbon (IC). After drying for 24 hours at 40 °C, both OC and $\delta^{13}C$ were measured using an elemental analyser coupled to an isotope ratio mass spectrometer (IRMS) at the NERC Life Science Mass Spectrometer Facility (Lancaster, UK). The standard deviation of

$\delta^{13}C$ triplicate measurements was 0.07 ‰ and $\delta^{13}C$ values are reported in standard delta notation relative to Vienna Pee Dee Belemnite (VPDB).

The samples in tin capsules were analysed for total carbon (TC) using a Elementar Elemental Analyser (EA) at the School of Geography and Sustainable Development, University of St Andrews (Verardo et al., 1990). Precision of the analysis is calculated based on repeat measurements of standard reference material B2178 (medium Organic content standard from Elemental Microanalysis, UK) with C = 0.08%.

These results were combined together to calculate the quantity of IC in each sample; the percentage of OC was subtracted from the percentage of TC to obtain the percentage of IC.

## 2.5 Binary mixing model

To discriminate between marine-sourced and terrestrially-sourced OC, we used a two end-members (binary) mixing model assuming a two-point source of OC, marine and terrestrial, following Thornton and McManus (1994). We used $\delta^{13}C$ measurements in the sediment samples as tracer of the source of OC in west Shetland voes and calculated the fraction of terrestrially-sourced OC ($OC_{terr}$) as follows:

$$OC_{terr} = (\delta^{13}C_{mar}-\delta^{13}C_{terr}) / (\delta^{13}C_{mar}-\delta^{13}C_{sample})$$

$$OC_{terr} + OC_{mar} = 1$$

where the end-member value for marine-sourced OC ($\delta^{13}C_{mar}$) was taken from Smeaton and Austin (2017) based on phytoplankton, zooplankton, macroalgae, and benthic microalgae carbon stable isotope values, while the end-member for terrestrially sourced OC ($\delta^{13}C_{terr}$) was based upon the work of Thornton et al. (2015) who studied the distribution of carbon stable isotope in Scotland's topsoil; $OC_{mar}$ indicates the fraction of marine-derived OC. We assumed no isotope discrimination between the source and the surface sediment samples.

## 2.6 Benthic foraminiferal counts and statistics

Aliquots of surface sediments were stained with Rose Bengal to allow the identification of living foraminifera, as detailed in Schönfeld et al. (2012). However, it should be noted that foraminiferal counts are 'total' (live + dead) because the main objective of this study is to provide a tool for the interpretation of fossil foraminiferal assemblages and their relationship with changes in OM and OC content in sediments over time (e.g.: Conradsen, 1993). Our priority is to understand how benthic foraminifera respond to prevailing (long-term) environmental conditions, rather than to seasonal variability. Additionally, scraping the top layer (~ 1 cm thick) of each grab with a domestic spoon may lead to a potential underestimation of living fauna if mixing of the top layer occurred during sampling. In this scenario, the number of living foraminifera at the sediment surface will be "diluted" due to the presence of dead/fossil specimens from deeper sediments. Having said this, a recent study by (Rillo et al., 2019) reported that historical sediment samples collected in a way that could have caused disturbance of the sediment surface (sounding and dredge) are still representative of surface conditions and their foraminiferal assemblages can

be used to reconstruct environmental changes reliably. At two sites, MD15-01 and MD15-05, replicate samples were taken (-01A and -01B, -05A and -05B, respectively) to check measurement reproducibility and seafloor heterogeneity.

As part of an unpublished work, the rose Bengal stained surface sediments were wet sieved over a 63 μm mesh; the residues oven-dried at < 60 °C, then weighed and dry-sieved into 63-150 μm and >150 μm size fractions. As part of an associated work aiming to understand the influence of different size fractions on the composition of benthic foraminiferal assemblages (Lo Giudice Cappelli et al., in review in *Frontiers*), both size fractions were analysed independently, and the results demonstrated that a more holistic picture of environmental change is obtained when benthic foraminiferal assemblages of both size fractions are compiled together. This improves the representation of the entire assemblage and results in more statistically robust environmental reconstructions. Similar findings were also reached by Weinkauf and Milker (2018 and references therein).

Depending on sample volume, each sample was divided into a number of splits using a standard dry-splitter and, when possible, at least 300 specimens were dry-picked to ensure statistical significance when discussing benthic foraminiferal assemblages' diversity and composition (Schönfeld et al., 2012). Species were identified following Austin (1991) and counted using a tally sheet; notes on tests preservations were made when signs of etching were observed.

Relative benthic foraminiferal abundances were calculated for each sample (Supplementary Table 1). It should be noted that we grouped under the name *E. excavatum* both forma *selseyense* and forma *clavata* despite them being recognised as genetically different species (Darling et al., 2016). It was not possible to consistently identify and discriminate between these two species, hence the use of a common identifier.

The Palaeontological Statistics software package PAST (version 3.16, Hammer et al., 2001) was used to analyse relationships between benthic foraminiferal assemblages through cluster analysis, canonical component analysis (CCA) and non-metric multidimensional scaling (MDS) based on the Bray-Curtis similarity index and relative abundance data. Cluster analysis showed good reproducibility of benthic foraminiferal assemblages (Fig. 4b) with the two pairs of replicate stations exhibiting assemblage reproducibility of 90% (MD15-01A and 01B) and 94% (MD15-05A and -05B).

## 3 Results

### 3.1 Sediment analyses

#### 3.1.1 Particle size distribution in west Shetland voes

A highly variable pattern is evident in the particle size distribution of the sediments of west Shetland voes (Fig. 3a). Clift Sound, Olna Firth, stations MD15-08 and-10 in Sand Sound, and station MD15-13 in Aith Voe have silt dominated sediments (>65%), (Fig. 3a). Conversely, stations MD15-06, -07,and -09 in Sand Sound, MD15-14 in Aith Voe, MD15-15 and -17 in Busta Voe, and stations MD15-20 and -21 in Vaila Sound are dominated by sandy sediments (>50%) (Fig.3a).

### 3.1.2 Organic matter content in west Shetland voes

Total OM is generally higher in proximity to land than in sites more exposed to the open sea (Fig. 3c). Site MD15-20 (Vaila Sound) has the lowest percentage of total OM (3.05%), ROM (2.24%) and LOM (0.81%), whereas site MD15-08 (Sand Sound) has the highest percentage of total OM (20.85%) and ROM (12.37%). LOM is the highest (8.81%) at site MD15-11 (Olna Firth) (Table 3; Fig. 3c).

### 3.1.3 Carbon content and source in west Shetland bulk sediments

The percentage of total carbon (TC) in west Shetland voes varies between 4.31 and 14.32% (Fig. 3d), with the highest percentage found at station MD15-05 (Clift Sound) and the lowest at station MD15-09 (Sand Sound). Organic carbon (OC) content ranges between 2.0-8.51%, peaking at station MD15-11 (Olna Firth) and reaching a minimum at site MD15-17 (Busta Voe) (Fig. 3d; Table 3). The percentage of inorganic carbon (IC) varies between and 0.35-7.51% and is the highest in station MD15-03 (Clift Sound) and the lowest in station MD15-14 (Aith Voe) (Fig. 3d). We excluded from the data set site MD15-15, as this sample contains coal, which is believed to have been shed accidentally from a commercial vessel. In general, there is a tendency for high OC and low IC at the head of the voes and the reverse occurs where the voes connect to the sea (Fig. 3d).

In terms of carbon source, $OC_{terr}$ is generally higher at sites close to land than in proximity to the open sea (Fig. 3b). It reaches a peak of 72% at station MD1-06 in Sand Sound, and a minimum of 21% at station MD15-03 (Fig. 3b).

### 3.1.4 Carbon stable isotopes ($\delta^{13}C$)

Bulk sediment $\delta^{13}C$ varies between -25.41 and -21.26 ‰ with an average value of -22.67 ‰ (Table 3). The lowest value (-25.41 ‰) is recorded at site MD15-06 (Sand Sound) and the highest (-21.26 ‰) at station MD15-03 (Clift Sound); high $\delta^{13}C$ values characterise marine dominated locations within each voe (Figs. 3c and 3d).

### 3.2 Benthic foraminiferal assemblages in west Shetland voes

We observed a total of 64 species of benthic foraminifera in the 21 stations analysed in this study, including the two replicates samples (Supplementary Table 1, Supplementary Fig. 2). Nine of the observed species are agglutinated: *Ammoscalaria runiana*, *E. scaber*, *Connemarella rudis*, *Haplophragmoides* sp., *Reophax fusiformis*, *Reophax scotii*, *Spiroplectinella wrightii*, *Textularia earlandii*, *Trochammina* sp. Of these, only *E. scaber* and *Trochammina* sp. were observed in all 21 stations (Supplementary Table 1). However, *E. scaber* relative abundance varies from 0.4 % in Vaila Sound (MD15-20) to 43.0 % in Sand Sound (MD15-10) (Fig. 5), whereas *Trochammina* sp. relative abundance reaches a maximum of 2.3 % in Clift Sound. Other common agglutinated species are *Haplophragmoides* sp. and *S. wrightii*, with relative abundance ranging from 0.1 %

(Vaila Sound) to 2.9 % (Busta Voe), and 0.1% (Olna Firth) and 3.9 % (Clift Sound), respectively (Supplementary Table 1). The distribution of the remaining agglutinated species is patchy, and their relative abundance varies around the average value of 1 % (Supplementary Table 1).

Six of the 64 observed species belong to the porcelaneous group Miliolina: *Cornuspira* sp. *Miliolinella subrotunda*, *Quinqueloculina bicornis*, *Quinqueloculina seminula*, *Quinqueloculina* sp., *Spiroloculina rotunda*. Of these, the most abundant species is *Q. seminula*, which is present in low numbers in five of the six studied fjords, it does not occur in Aith Voe. The general distribution of Miliolina is patchy and their relative abundance varies around the average value of 0.5 %, with the highest value of 2.4 % observed at the mouth of Clift Sound (MD15-05B) (Supplementary Table 1).

The remaining 49 species are hyaline foraminifera of which 9 species have relative abundances of > 10 % in at least one of the 21 stations studied: *Ammonia* spp., *Bulimina marginata*, *Buliminella elegantissima*, *Cibicides* spp., *Elphidium gerthi*, *Elphidium margaritaceum*, *Elphidium excavatum*, *Rosalina* spp. and *Stainforthia fusiformis* (Table 2). Additionally, of all the hyaline species, 20 are rare having relative abundances < 1 %, while 18 species are frequent having a relative abundances between 1-5 % and only 2 species are common having relative abundances ranging between 5-10 % (Supplementary Table 1, Supplementary Fig. 2).

Overall, ten taxa of benthic foraminifera were observed in each one of the six voes and had relative abundances of more than 10 % in at least one of the 21 stations studied (Table 2): *Ammonia* spp., *Bulimina marginata*, *Buliminella elegantissima*, *Cibicides* spp., *Eggerelloides scaber*, *Elphidium gerthi*, *Elphidium margaritaceum*, *Elphidium excavatum*, *Rosalina* spp. and *Stainforthia fusiformis*. These ten taxa were categorised as dominant and imaged with a scanning electron microscope (SEM) at the Scottish Oceans Institute, University of St Andrews (Fig. 5). Altogether, these ten dominant taxa account for more than 70% of the total assemblage at each station (Table 2). In general, the most abundant taxa observed in west Shetland's six voes is the hyaline *Cibicides* spp. relative abundance, followed by the agglutinated *E. scaber* relative abundance (Table 2 and Fig. 5).

### 3.2.1 Statistical analyses of benthic foraminifera data and environmental parameters

We used cluster analysis of environmental parameters (BWT, BWS, O$_2$, % Clay, OC, IC, ROM, LOM, $\delta^{13}$C and WD) based on Euclidean distance and of benthic foraminifera relative abundance based on the Bray-Curtis index to illustrate similarities between locations (Figs. 4a and 4b), and performed CCA on the combined dataset (benthic foraminifera + environmental parameters) to examine the response of benthic foraminiferal assemblages to environmental gradients (Fig. 4c). Additionally, we run non-metric MDS based on the Bray-Curtis similarity index and foraminifera relative abundance data to analyse for similarities between the populations of the 21 studied stations, independently from environmental forcing (Fig. 5). To note that temperature, salinity and oxygen measurements are from 2009 and may not fully represent the conditions in 2015 when the sediment samples were collected.

Based on both CCA and non-metric MDS, we defined the same four groups with 75% confidence; in the CCA plot axis 1 explains 57 % of total variability and axis 2 the 26 % based on the 10 environmental parameters (Figs. 4c and 5). Group 1 includes assemblages dominated by the genus *Elphidium*, and by *E. scaber*; the head of Clift Sound falls into this cluster (MD15-06 to -09). Group 2 contains assemblages dominated by *E. scaber* with *S. fusiformis* and *B. marginata* as associated species; the inner basin of Sand Sound (MD15-10), Olna Firth (MD15-11 and -12), and MD15-16 in Busta Voe are part of this cluster. Group 3 consists of assemblages characterised by relatively high diversity and no obvious dominant species; MD15-01A and -01B (head of Clift Sound), MD15-13 and -14 (Aith Voe); MD15-15 (Busta Voe); MD15-18 and -19 (head of Vaila Sound) belong to this group. Group 4 comprises assemblages dominated by the genus *Cibicides* with *Rosalina* as associated taxa and includes stations MD15-02 to MD15-05 in Clift Sound and stations MD15-20 and -21 in Vaila Sound (Figs. 4c and 5). Station MD15-17 in Busta Voe falls just outside the 75% confidence interval defining Group 4; however, cluster analysis (Fig. 4b) shows ~82% similarity between the assemblage of this site and station MD15-20 and more than the 70% similarity with the rest of the locations belonging to Group 4. Therefore, for the purposes of our discussion, we include station MD15-17 in Group 4.

## 4 Discussions

### 4.1 Particle size distribution in west Shetland voes

The particle size data from the voes of Shetland exhibit a large range with the silt and sand fractions, but little variation in the proportion of clay (Fig. 3a). These samples suggest a significant size sorting which is consistent with the tidal and estuarine dynamics of these voes. In Vaila Sound, intensified bottom currents and winnowing of finer sediments owing the influx of Atlantic Ocean waters (Cefas, 2009) appear to control local grain size distribution (Fig. 3a). Here, stations exposed to the open sea and intensified bottom currents are characterised by coarser sediments (MD15-20 and 21), whereas more sheltered and calm locations (MD15-18 and -19) are characterised by finer sediments (Fig. 3a). In contrast, sediments in Clift Sound are silt-dominated (Fig. 3a) despite the unrestricted geomorphology (Fig. 1) and vigorous bottom currents (Cefas, 2007a). It appears that a different mechanism than winnowing of finer sediments by intensified bottom currents drives particle size distribution here. In Clift Sound, the overall low percentage of OM together with the relatively high IC content and $\delta^{13}$C values point to a reduction in terrigenous input as the main driver of particle size distribution (Fig. 3). Conversely, in Sand Sound, an increased terrigenous input appears to be responsible for the coarser sediments found in stations MD15-06, -07 and -09 (Fig. 3). The head of the voe is a sheltered area that receives freshwater from a significant number of rivers draining the surrounding land (Cefas, 2007b, 2008b), resulting in an overall high percentage of OM and OC, low $\delta^{13}$C values (Figs. 3c and 3d) and coarser sediments (Fig. 3a). In contrast, station MD15-08 at the head of Sand Sound is characterised by finer sediments (Fig. 3a) despite the high terrigenous supply at this location (Figs. 3c and 3d). It is possible that, due to the restricted geomorphology of Sand Sound (Fig. 1), weak currents are unable to transport the coarse material to station MD15-08, the deepest station at the

head of the voe (11 m vs. 6 m). In Sand Sound inner basin (MD15-10), silty sediments dominate (Fig. 3a). Here, a combination of reduced terrigenous input and/or weaker currents could be driving particle size distribution. Nørgaard-Pedersen et al. (2006) found similar drivers of particle size distribution in Loch Etive, a sea loch in mainland western Scotland. Vigorous bottom currents and/or enhanced terrigenous supply would result in locally coarse sediments, whereas weaker currents and/or limited

terrigenous supply would result in finer sediments. Due to the duality of drivers determining particle size distribution in west Shetland voes (bottom currents intensity vs. terrigenous supply), it is challenging to consistently group locations based on granulometry data (Fig. 3a) or identify a relationship between grain size distribution, OM and OC content in the sediments (Supplementary Fig. 3).

## 4.2 Spatial distribution, quantity, quality and source of organic matter, and carbon in west Shetland voes.

A clear pattern is evident with regard to the quantity, quality and source of organic matter and carbon measured in the six west Shetland voes (Fig. 3). Typically, carbon is present in the sediments as IC and OC and the distribution of these two forms of carbon tends to mirror one another, with low IC and high OC percentages towards land while the reverse occurs close to the open sea (Fig. 3d). Organic matter follows the same pattern of OC distribution regardless of its quality (refractory vs. labile); however, ROM content in the sediments is always higher than LOM (Fig. 3c) owing its resilience to degradation.

About the source of OM and OC, low $\delta^{13}C$ and high $OC_{terr}$ values characterise the head of the voes where high ROM and OC are accumulated, indicating that the source of this material is terrestrial (Fig. 3). Terrestrial OM is more stable and resistant to decay than marine OM (Batten, 1996) and it is enriched in $^{12}C$ from soils resulting in lower $\delta^{13}C$ values than marine OM derived from algae (e.g. Balasse et al., 2005; Ficken et al., 1998; Marconi et al., 2011; Schiener et al., 2014; Schmidt and Gleixner, 2005). Sand Sound is dominated by organic rich and terrestrially sourced material as it receives freshwater and

terrestrial OM from the numerous streams draining the surrounding land (Fig. 3). In contrast, Clift and Vaila Sounds are characterised by marine sourced material and high IC content in their sediments (Fig. 3) as the inflow of Atlantic Ocean waters improves the voes' potential for IC storage (Burrows et al., 2017; Smeaton et al., 2016, 2017). Olna Firth has the highest LOM percentage as this relatively deep and micro-tidal basin (Cefas, 2013) is prone to water mass stratification which can lower bottom water oxygenation and slow down the reworking/remineralisation of OM favouring the preservation of the more labile

material (Fig. 3).

A slight deviation from this overall trend was recorded at station MD15-05. We speculate that the unusually high TC content at station MD15-05 at the mouth of Clift Sound (Fig. 3d) could relate to the MV Braer oil spill of 1993. It appears that after the spill, due to the adverse weather, the oil became deposited in the sediment of Burra Haaf, just west of Clift Sound (Fisheries Research Services). Despite the fact that oil levels in the contaminated sediments have been decreasing steadily over the

intervening years, it is plausible that the OC measurements in this study are detecting remnants of this event. Nevertheless, the benthic foraminiferal assemblage of this location is dominated by *Cibicides* spp. (see par. 4.3 for more details) which does not

reflect the oil spill nor the high OC content, leaving an open question regarding what drives OC deposition and accumulation at the mouth of Clift Sound.

## 4.3 Modern benthic foraminiferal distribution in Shetland.

### 4.3.1 Group 1 – *Elphidium* spp. and *E. scaber*

The genus *Elphidium* becomes the dominant taxa (relative abundance ~38%) at the head of Sand Sound (Fig. 5), a very shallow (max depth 11m) and organic rich basin (OC 5.2%, ROM 7.3% and LOM 4.9%; Fig. 4c). Specifically, *E. excavatum* is the most abundant *Elphidium* (~21%, Fig. 5), as it can live close to freshwater outflows, in low energy environments, and tolerates high OC and OM concentrations (Fig. 4c) (Alve and Murray, 1999; Jennings, 2004; Korsun et al., 2014; Mendes et al., 2012). This is consistent with streams draining into the head of Sand Sound transporting terrestrial and organic rich material into the

voe (Figs. 3 and 4c). Additionally, the high relative abundance of *E. scaber* (26%) at the same locations (Fig. 5) purports a low energy environment, possible water mass stratification and high ROM (Fig. 4c) (Evans et al., 2002; Fontanier et al., 2002; Murray, 1992; Nørgaard-Pedersen et al., 2006). Thus, assemblages dominated by *Elphidium* spp. and *E. scaber* are likely to be found in land-locked regions influenced by riverine/freshwater inputs like the head of fjords with a restricted geomorphology.

### 4.3.2 Group 2 – *E. scaber*, *B. marginata* and *S. fusiformis*

*Eggerelloides scaber*, *B. marginata* and *S. fusiformis* are species typical of organic rich sediments, stratified waters and possibly low oxygen concentrations (Fig. 4c), often suggesting stressed environments (Alve, 1994, 2003; Evans et al., 2002; Fontanier et al., 2002; Murray, 1992; Nørgaard-Pedersen et al., 2006). *Bulimina marginata* (~9%) seems to thrive in regions with high ROM (9.6%) due to its ability to use OM of low nutritional value as food (Klitgaard-Kristensen & Buhl-Mortensen

1999) (Figs. 4c and 5). *Stainforthia fusiformis* (~21%) is a known opportunistic species which follows high OM gradients at stratification fronts (Alve, 2003; Alve and Murray, 1997; Scott et al., 2003). In general, high organics in the sediments can result in low oxygen concentrations at the seafloor (Fig. 4c) due to eutrophication and increased respiration (Bianchi et al., 2016). Additionally, the production of humic acids as a by-product of organic matter biodegradation may lower the pH (Bauer and Bianchi, 2012), overall making the environment more hostile for foraminifera to calcify, which could result in the

dominance of agglutinated species like *E. scaber* (~31%) (Fig. 5). Etching of *Ammonia* spp. shells (Supplementary Fig. 2) in Olna Firth supports local acidification of bottom waters, which is substantiated by the low IC content (2.6%) of stations MD15-11 and -12 (Fig. 3). Therefore, assemblages characterised by *E. scaber*, *B. marginata* and *S. fusiformis* reflect a stressed environment with a heavily stratified water column - possibly low oxygen levels, and organic rich sediment especially in ROM of low nutritional value. Despite seawater properties (T, S and $O_2$) were measured in 2009, they seem to be representative of

seawater characteristic in 2015 when the sediment samples were collected, as the two sets of measurements (seawater and sediments) appear to be in agreement, reflecting long-term of environmental dynamics.

### 4.3.3 Group 3 – no obvious dominant taxa

The assemblage of Group 3 is characterised by high relative abundance of taxa with affinities for OC (4.9%) and OM (6.3%) like *Elphidium* (8%)*, E. scaber* (13%) and *B. elegantissima* (5%) (e.g.: Mendes *et al.* 2012; Alve & Murray 1999) as well as high energy taxa like *Cibicides* (22%) and *Ammonia* (5%) (Scott et al., 2003) (Figs. 4c and 5). The low relative abundance of *B. marginata* (2%) and *S. fusiformis* (5%) additionally suggests that OM at these locations has good nutritional value and current activity is present at these sites (reduced stratification, possibly vertical mixing). Therefore, assemblages showing this composition are likely to reflect depositional environments with mild or episodic current activity and moderate organic content.

### 4.3.4 Group 4 – *Cibicides* spp. and *Rosalina* spp.

In Group 4, the dominant taxa is *Cibicides* (~57%) followed by *Rosalina* (8%) (Fig. 5), which reflect marginal to open sea conditions and vigorous bottom currents (Fig. 4c) (Alve & Murray 1999; Jennings 2004; Klitgaard-Kristensen & Sejrup 2002). A clear relationship is evident between low organics (OC 4.2%, ROM 3.2% LOM 1.5%) (Fig. 4c) and the dominance of *Cibicides* (Figs. 4c and 5). All locations falling in Group 4 are on the path of Atlantic Ocean waters entering the fjords meaning that this assemblage composition is most likely found in areas dominated by sea processes.

### 4.3.5 Other drivers influencing modern benthic foraminifera distribution

Seawater properties can compete with OC and OM in influencing the distribution of benthic foraminifera across the six voes, especially BWS and $O_2$ concentration may exert a second-order control on the composition of foraminiferal assemblages (Jorissen et al., 1995; Van Der Zwaan et al., 1999). A highly variable BWS generally influences the distribution of stenohaline (strictly marine species) and euryhaline species (able to adapt to a wide range of salinity) along BWS gradients; however, in Shetland, BWS is almost constant across the voes (Fig. 2) having very minimal influence on benthic foraminiferal assemblages (Fig. 4c). Similarly, bottom waters are mostly well-oxygenated ($[O_2] > 5$ mg l$^{-1}$) in all the voes (Fig. 2), meaning that food availability is the main driver of changes in the composition of benthic assemblages (Jorissen et al., 1995; Van Der Zwaan et al., 1999). It should be noted that seawater properties reported in Fig. 2 were measured in 2009 and may differ from the actual conditions in 2015 when sediment samples were collected; nevertheless, our results show that similar dynamics were most likely in place in 2015, as the two sets of measurements compare very well in depicting local environmental dynamics. Food availability is deeply connected to organic fluxes and transport of organic matter to the seafloor, hence the relationships found between benthic foraminiferal assemblages and depositional environments (Figs. 4c and 5). Only one station in Olna Firth (MD15-11) has poorly oxygenated bottom waters ($O_2 < 5$ mg l$^{-1}$), which could represent the onset of hypoxic conditions (Vaquer-Sunyer and Duarte, 2008) potentially affecting the distribution of benthic foraminifera (Fig. 2). The benthic

foraminiferal assemblage of this location reflects not only organic rich sediments, but also stratified waters and low oxygen concentrations, pointing to a stressed environment (Fig. 4c). However, this is far from the dysoxic/anoxic conditions required for a switch from food-limited to oxygen-limited conditions (Fig. 2). Therefore, the quantity and quality (labile vs. refractory) of organic matter reaching the seafloor is still the major control on the benthic foraminiferal assemblage found at station
MD15-11.

### 4.4 Comparison with other fjords.

### 4.4.1 Spatial distribution and sources of OM and C.

The processes governing OM and C inputs in west Shetland voes are comparable with those reported for other Scottish fjords by Smeaton et al. (2017) who found a relationship between the OC content in Scottish fjords and their physical characteristics
such as tidal range, precipitation, catchment area and runoff. However, as catchments in west Shetland transport sediment from the adjacent C rich peatlands into the marine system, west Shetland voes overall store more TC (Supplementary Fig. 1) compared to other Scottish fjords (Smeaton et al., 2016). On average, the TC content of the six west Shetland voes is 8.8%, whereas TC in other Scottish fjords ranges between 0-8% (Smeaton et al., 2017 in Fig. 5). Nevertheless, as stated in Smeaton et al. (2017), there are similarities between Scotland's mainland fjords and these six voes in west Shetland. For example, fjords
like Loch Broom and Little Loch Broom in Scotland have relatively unrestricted geomorphologies which makes them highly efficient in capturing IC, similar to Clift Sound and Vaila Sound in Shetland (Fig. 3). Transient low oxygen concentrations in Scotland's mainland fjords seem to be an important factor in the burial and preservation of OC (Gillibrand et al., 2006). Similarly, low oxygen levels in Olna Firth (Fig. 2), also indicated by benthic foraminiferal assemblages (Fig. 4c), are associated with the preservation of labile organic matter and OC (Figs. 3 and 4c). Additionally, carbon stable isotope values in Shetland
are in range with carbon stable isotope measurements of other Scottish sea lochs: Sunart (average -22.37 ‰, ranging from -24.42 to -21.37 ‰), Teacius (average -24.79 ‰, n=15) (Smeaton and Austin, 2017), Creran (average -21.98 ‰ varying between -24.7 to -17.3 ‰), Etive (average -25.8 ‰, n=3) (Loh et al., 2008).

### 4.4.2 Benthic foraminiferal assemblages

Our knowledge of benthic foraminiferal assemblages in Scottish fjords is limited to studies based on fauna from Loch Etive
(Murray et al., 2003; Nørgaard-Pedersen et al., 2006) and their response to changes in sea level and terrigenous input entering the catchment. Species in common with west Shetland voes include *Cibicides lobatulus*, *Ammonia batavus* and *E. scaber*. Assemblages dominated by *C. lobatulus* and *A. batavus* in Loch Etive indicate marine conditions and intensified bottom currents possibly associated with deep water renewal events (Nørgaard-Pedersen et al., 2006). This agrees well with the assemblages dominated by *Cibicides* spp. and *Rosalina* spp. observed in Clift Sound and Vaila Sound suggestive of high-
energy environments (Fig. 5). The dominance of agglutinated species such as *E. scaber* occurs in Loch Etive when OM

deposition increases eventually resulting in low oxygen levels in bottom waters and carbonate dissolution (Murray et al., 2003; Nørgaard-Pedersen et al., 2006). We observed similar environmental conditions and assemblages in Olna Firth, the middle of Busta Voe and inner basin of Sand Sound (Figs. 4c and 5).

Similarities can also be observed between assemblages from the Scottish coast and shelf and west Shetland voes. Klitgaard-
Kristensen and Sejrup (2002) and Murray (2003) found high abundance of *C. lobatulus* and *Rosalina* sp. in areas with strong currents, low organics and coarse-grain sediments. We do not see a correlation between grain size and *Cibicides* or *Rosalina* in our samples due to the multiple forcings driving particle size distribution in Shetland (bottom current intensity vs. terrigenous supply); however we do observe a clear correlation between high relative abundance of *Cibicides* and *Rosalina* and low organics and high energy environments (Figs.4c and 5). Klitgaard-Kristensen and Sejrup (2002) and Murray (2003) also found
a relationship between the dominance of *B. marginata* and low oxygen levels, high organics and fine sediments. Additionally, Murray (2003) also observed *S. fusiformis* dominated assemblages in association with high availability of LOM and low oxygen levels. Overall, this compares really well with the assemblage we described in Group 2 (Figs. 4c and 5).

A recent study by Murray and Alve (2016) synthesise benthic foraminiferal biogeography in NW European fjords, mostly focusing on Norwegian fauna. Similarities with west Shetland voes include *C. lobatulus* as an "Atlantic species" pointing to
the influence of Atlantic Ocean waters on local estuarine/fjordic circulation. Assemblages dominated by *E. scaber* have been observed in Oslofjord and Drammensfjord in Norway (Alve, 1995; Alve and Nagy, 1986) and Gullmar Fjord in Sweden (Qvale et al., 1984), generally associated with organic rich sediments and shallow waters. Additionally, Klitgaard-Kristensen and Buhl-Mortensen (1999) found high relative abundance of the opportunistic species *S. fusiformis* in Norwegian fjords with a high per cent of sand and organic carbon, and assemblages characterised by *B. marginata* in environments rich in ROM.

In our sample set, we had no means to distinguish *in situ* tests from potentially advected tests when picking and counting unstained specimens, and the close match between our benthic foraminiferal assemblages and published data suggests that advection of material from other locations, especially in the more energetic environments, is negligible. Overall, benthic foraminiferal assemblages do respond to variations in the amount of OM and OC becoming deposited in fjord sediments (Fig. 4c), and comparisons can be drawn between fjordic systems. We recognise that this is a qualitative approach and we are not
able to quantify changes in OM or OC content reaching the seafloor based on foraminiferal assemblages. However, this study provides an initial insight into the use of foraminifera to discriminate between post-depositional OC degradation and actual OC burial and accumulation in fjords, as foraminifera would only preserve the latter information in their assemblage composition. We reckon however that extreme conditions at the seafloor may result in post depositional dissolution of benthic foraminiferal tests, which would obviously compromise the assemblage composition. For this reason, we recommend to
carefully consider the level of preservation of each assemblage and, when necessary, make appropriate remarks on etching of tests and other possible signs of tests dissolution (e.g. Khanna et al., 2013). Additionally, assemblages dominated by agglutinated species like in Group 2 might have a lower diagnostic potential in their application down-core compared to their hyaline and porcelaneous counterpart due to a tendency of being poorly preserved in the fossil record. Nevertheless, foraminifera are generally abundant in the fossil record and this approach shows down-core potential to support reconstructions

of changes in OC burial and accumulation over time and could be paired with statistical and spatial model studies to tease out post depositional OC degradation.

## 5 Conclusions

Our records from west Shetland voes provide new insights into sediments provenance and type, modern benthic foraminiferal spatial distribution across environmental gradients, and their use as indicators for OC enrichment in fjordic systems. Our results support a terrestrial source of both OM and OC in land-locked areas and calm depositional settings in west Shetland voes. Additionally, in locations characterised by low bottom water oxygenation, LOM is present in higher percentages than in well ventilated sites. Conversely, marine processes govern IC storage, with higher concentrations in marginal to open marine settings. Benthic foraminiferal assemblages reliably reflect changes in OC content reaching the seafloor and show great potential as bio-indicators to trace these changes over time. We were able to identify four groups (75% confidence) based on foraminiferal relative abundance data, which reflect four types of depositional settings: 1) Land-locked with high organics and very weak currents; 2) Stressed environment with high organics, heavily stratified water column and low oxygen concentrations; 3) Transitional environment characterised by moderate concentrations of organics and mild or intermittent currents; 4) Marginal to coastal settings with low organics and a high-energy environment.

Therefore, studies aiming to reconstruct past variations in the amount of OC reaching the seafloor and becoming stored therein could benefit from the use of benthic foraminifera assemblages as a qualitative approach to understand OC preservation in the sediment column and, for example, ground truth model predictions.

*Author contribution*. WENA and ELGC designed the study. ELGC, JC and CS analysed sediment physical properties, while ELGC and WENA worked on benthic foraminiferal assemblages. KD provided hydrographic data. ELGC prepared the manuscript with contributions from all co-authors.

*Competing interests*. The authors declare that they have no conflict of interest.

*Acknowledgements*. The crew of the MV Moder Dy are gratefully acknowledge for support of the work during the cruise in Shetland. We are thankful to undergraduate students Laura Smith, Leonie Strobl and Bethan Hudd for their help in the microscopy lab assisting with benthic foraminifera picking. This project was supported by BBSRC/NERC (ref. BB/M026620/01). Isotope analysis was supported by NERC Life Science Mass Spectrometry Facility (CEH_L_098_11_2015). KD was funded by the EU INTERREG IV Northern Periphery project: Warning of Algal Toxin Events to support Aquaculture in the NPP coastal zone Region "WATER". We thank two anonymous referees and the Handling Associate Editor Dr Hiroshi Kitazato for valuable and constructive comments that improved the manuscript.

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

**Figures Captions**

**Figure 1: Location Map of Shetland and sampling stations.**

**Figure 2: CTD profiles from 2009 survey**. a) Temperature profiles and bottom water tempreature (BWT) at each sampling station. b)
Salinity profiles and bottom water salinity (BWS) at each sampling station. c) Dissolved oxygen profiles and oxygen concentration in bottom waters at each sampling station. Dashed line at 5 mg l$^{-1}$ indicate possible onset of hypoxic conditions.

**Figure 3: Sediment physical properties.** a) Particle size analysis. b) Fraction of terrestrially-source OC (OC$_{terr}$)and marine-source OC (OC$_{mar}$) at each site. c) Refractory and labile organic matter content at each site and carbon stable isotope. d) Inorganic and organic carbon percentage at each site and carbon stable isotope.

**Figure 4: Statistical analyses.** a) Cluster analysis of sediment and seawater physical properties. b) Cluster analysis of benthic foraminiferal assemblages. c) Canonical component analysis of the combined datasets (sediment and seawater + benthic foraminifera relative abundance).

**Figure 5: Non metric MDS-ordination plot** of benthic foraminifera assemblages from each site based on relative abundance data.

Table 1. Station Data

| Location | Sample ID | Longitude | Latitude |
|---|---|---|---|
| Clift Sound | MD15-01 | -1.277267 | 60.11943 |
| Clift Sound | MD15-02 | -1.284533 | 60.09987 |
| Clift Sound | MD15-03 | -1.2975 | 60.07995 |
| Clift Sound | MD15-04 | -1.307783 | 60.06635 |
| Clift Sound | MD15-05 | -1.315883 | 60.04765 |
| Sand Sound | MD15-06 | -1.40295 | 60.24485 |
| Sand Sound | MD15-07 | -1.404183 | 60.24477 |
| Sand Sound | MD15-08 | -1.379517 | 60.24237 |
| Sand Sound | MD15-09 | -1.35835 | 60.24178 |
| Sand Sound | MD15-10 | -1.370517 | 60.22822 |
| Olna Firth | MD15-11 | -1.295883 | 60.36277 |
| Olna Firth | MD15-12 | -1.32315 | 60.36425 |
| Aith Voe | MD15-13 | -1.374883 | 60.28945 |
| Aith Voe | MD15-14 | -1.373333 | 60.3055 |
| Busta Voe | MD15-15 | -1.361217 | 60.39075 |
| Busta Voe | MD15-16 | -1.358883 | 60.38063 |
| Busta Voe | MD15-17 | -1.370517 | 60.3694 |
| Vaila Sound | MD15-18 | -1.564083 | 60.21943 |
| Vaila Sound | MD15-19 | -1.5624 | 60.213 |
| Vaila Sound | MD15-20 | -1.582167 | 60.21428 |
| Vaila Sound | MD15-21 | -1.567467 | 60.20567 |

Table 2. Relative abundance of the ten dominant taxa.

| Location | Sample ID | *Ammonia* spp. | *B. marginata* | *B. elegantissima* | *Cibicides* spp. | *E. scaber* | *E. gerthi* | *E. margaritaceum* | *E. excavatum* | *Rosalina* spp. | *S. fusiformis* | Total % |
|---|---|---|---|---|---|---|---|---|---|---|---|---|
| Clift Sound | MD15-01A | 0.2 | 0.5 | 2.1 | 14.5 | 22.0 | 10.7 | 20.1 | 7.8 | 3.8 | 3.0 | **84.6** |
| Clift Sound | MD15-01B | 0.2 | 0.6 | 2.1 | 15.0 | 19.9 | 10.7 | 24.3 | 2.1 | 3.2 | 4.4 | **82.5** |
| Clift Sound | MD15-02 | | 0.7 | 0.3 | 47.6 | 10.6 | 5.4 | 3.1 | 1.9 | 6.6 | 3.8 | **80.3** |
| Clift Sound | MD15-03 | 0.2 | 1.2 | 0.7 | 53.7 | 3.3 | 10.5 | 0.8 | 1.2 | 5.4 | 4.6 | **81.5** |
| Clift Sound | MD15-04 | 0.3 | 0.9 | 0.3 | 48.9 | 5.2 | 13.9 | 0.2 | | 4.7 | 2.9 | **77.3** |
| Clift Sound | MD15-05A | | 0.3 | | 57.6 | 2.1 | 0.4 | 0.1 | | 9.7 | 0.7 | **71.0** |
| Clift Sound | MD15-05B | | 0.6 | | 56.7 | 2.1 | 0.2 | 0.0 | 0.5 | 10.4 | 1.5 | **72.0** |
| Sand Sound | MD15-06 | 4.8 | 0.5 | 4.6 | 2.0 | 25.5 | 15.6 | 6.1 | 15.3 | | 1.3 | **75.7** |
| Sand Sound | MD15-07 | 6.5 | 0.8 | 4.8 | 5.5 | 27.7 | 11.7 | 7.6 | 21.5 | 0.7 | | **86.7** |
| Sand Sound | MD15-08 | 3.0 | 1.7 | 11.4 | 2.0 | 29.7 | 7.5 | 1.4 | 22.4 | 0.2 | 6.3 | **85.6** |
| Sand Sound | MD15-09 | 5.4 | 0.9 | 8.8 | 1.9 | 19.7 | 9.8 | 10.1 | 25.6 | 0.2 | 1.6 | **83.9** |
| Sand Sound | MD15-10 | 0.9 | 3.8 | 1.9 | 8.3 | 43.0 | 4.7 | 1.3 | 5.2 | 1.4 | 11.3 | **81.8** |
| Olna Firth | MD15-11 | | 14.8 | 0.8 | 10.9 | 31.9 | 4.4 | 1.8 | 0.8 | 0.5 | 23.1 | **88.8** |
| Olna Firth | MD15-12 | 8.5 | 8.6 | 0.7 | 13.7 | 22.9 | 2.2 | 0.9 | 2.4 | 0.3 | 26.0 | **86.2** |
| Aith Voe | MD15-13 | 3.0 | 5.6 | 12.6 | 14.2 | 11.7 | 9.3 | 4.0 | 10.0 | 0.3 | 7.3 | **78.1** |
| Aith Voe | MD15-14 | 12.6 | 4.1 | 4.6 | 26.1 | 10.0 | 8.2 | 0.9 | 4.3 | 1.8 | 8.2 | **80.8** |
| Busta Voe | MD15-15 | 1.0 | 2.8 | 2.0 | 30.1 | 12.9 | 7.0 | 5.1 | 5.6 | 0.8 | 8.2 | **75.5** |
| Busta Voe | MD15-16 | 5.7 | 8.5 | 0.9 | 7.6 | 26.4 | 5.0 | 1.2 | 5.1 | 0.2 | 23.1 | **83.7** |
| Busta Voe | MD15-17 | 2.2 | 1.2 | 0.1 | 70.1 | 2.8 | 1.4 | 0.8 | 1.5 | 1.1 | 4.8 | **86.1** |
| Vaila Sound | MD15-18 | 6.1 | 0.9 | 11.5 | 24.4 | 9.5 | 7.6 | 9.0 | 9.2 | 2.3 | 4.0 | **84.5** |
| Vaila Sound | MD15-19 | 10.2 | 2.1 | 1.8 | 29.6 | 8.8 | 8.3 | 3.4 | 9.2 | 4.5 | 4.3 | **82.3** |
| Vaila Sound | MD15-20 | | 0.4 | 0.1 | 68.7 | 0.4 | 1.2 | 0.9 | 1.2 | 14.7 | 0.5 | **88.2** |
| Vaila Sound | MD15-21 | 0.3 | 1.4 | 0.3 | 56.8 | 1.9 | 1.9 | 1.0 | 2.9 | 11.3 | 1.4 | **78.9** |

Table 3. Sediment physical properties

| Location | Sample ID | Water depth (m) | $\delta^{13}C$ | %IC | %OC | %LOM | %ROM | %OC$_{terr}$ |
|---|---|---|---|---|---|---|---|---|
| Clift Sound | MD15-01A | 13 | -23.16 | 3.55 | 5.44 | 4.35 | 6.99 | 45 |
| Clift Sound | MD15-01B | 13 | -23.16 | 3.55 | 5.44 | 3.94 | 7.29 | 45 |
| Clift Sound | MD15-02 | 15 | -21.92 | 7.07 | 2.80 | 1.59 | 2.95 | 30 |
| Clift Sound | MD15-03 | 23 | -21.26 | 7.51 | 3.48 | 1.60 | 3.38 | 21 |
| Clift Sound | MD15-04 | 21 | -21.51 | 4.72 | 3.58 | 1.78 | 3.79 | 25 |
| Clift Sound | MD15-05A | 20 | -21.35 | 7.12 | 7.20 | 1.32 | 4.47 | 23 |
| Clift Sound | MD15-05B | 20 | -21.35 | 7.12 | 7.20 | 1.40 | 4.09 | 23 |
| Sand Sound | MD15-06 | 6 | -25.41 | 2.41 | 4.33 | 5.32 | 8.50 | 72 |
| Sand Sound | MD15-07 | 6 | -24.07 | 0.53 | 4.51 | 2.63 | 4.26 | 56 |
| Sand Sound | MD15-08 | 11 | -23.84 | 1.67 | 8.28 | 8.48 | 12.37 | 53 |
| Sand Sound | MD15-09 | 6 | -23.64 | 0.53 | 3.78 | 3.07 | 4.24 | 50 |
| Sand Sound | MD15-10 | 21 | -22.87 | 5.57 | 4.80 | 4.92 | 6.68 | 41 |
| Olna Firth | MD15-11 | 31 | -22.36 | 3.02 | 8.51 | 8.81 | 10.89 | 35 |
| Olna Firth | MD15-12 | 35 | -22.65 | 2.15 | 7.54 | 8.07 | 11.83 | 38 |
| Aith Voe | MD15-13 | 12 | -23.55 | 4.70 | 5.44 | 5.54 | 7.07 | 49 |
| Aith Voe | MD15-14 | 36 | -23.38 | 0.35 | 4.48 | 2.06 | 3.92 | 47 |
| Busta Voe | MD15-15 | 18 | - | - | - | 2.99 | 8.27 | - |
| Busta Voe | MD15-16 | 26 | -23.06 | 7.40 | 3.27 | 5.48 | 9.19 | 43 |
| Busta Voe | MD15-17 | 24 | -22.59 | 5.47 | 2.06 | 2.54 | 3.93 | 38 |
| Vaila Sound | MD15-18 | 14 | -21.82 | 3.70 | 4.79 | 3.03 | 6.00 | 28 |
| Vaila Sound | MD15-19 | 21 | -21.54 | 3.91 | 3.51 | 2.86 | 6.38 | 25 |
| Vaila Sound | MD15-20 | 19 | -21.6 | 4.07 | 4.07 | 0.81 | 2.24 | 26 |
| Vaila Sound | MD15-21 | 21 | -21.75 | 5.52 | 2.89 | 1.16 | 3.03 | 27 |

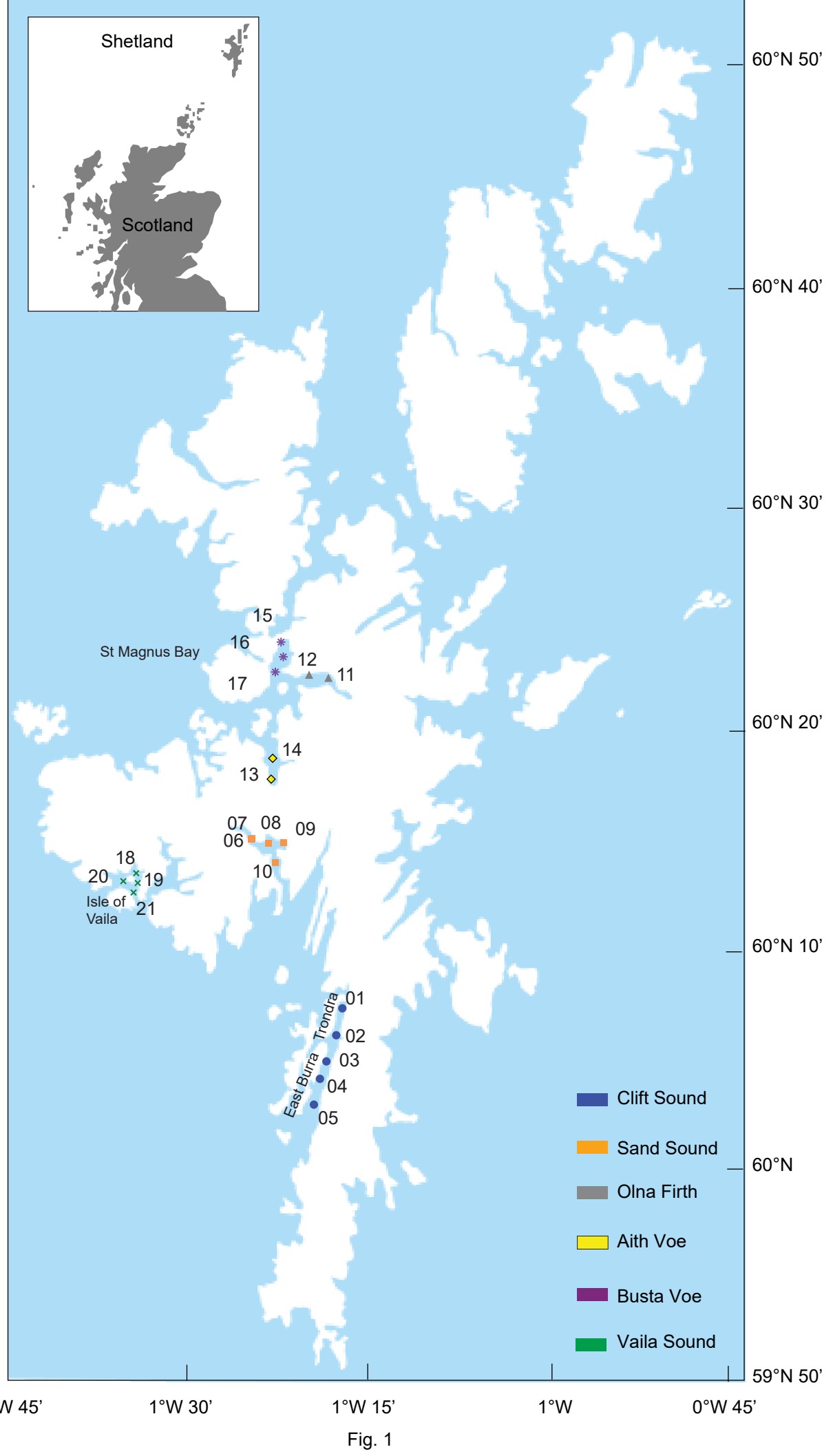

Fig. 1

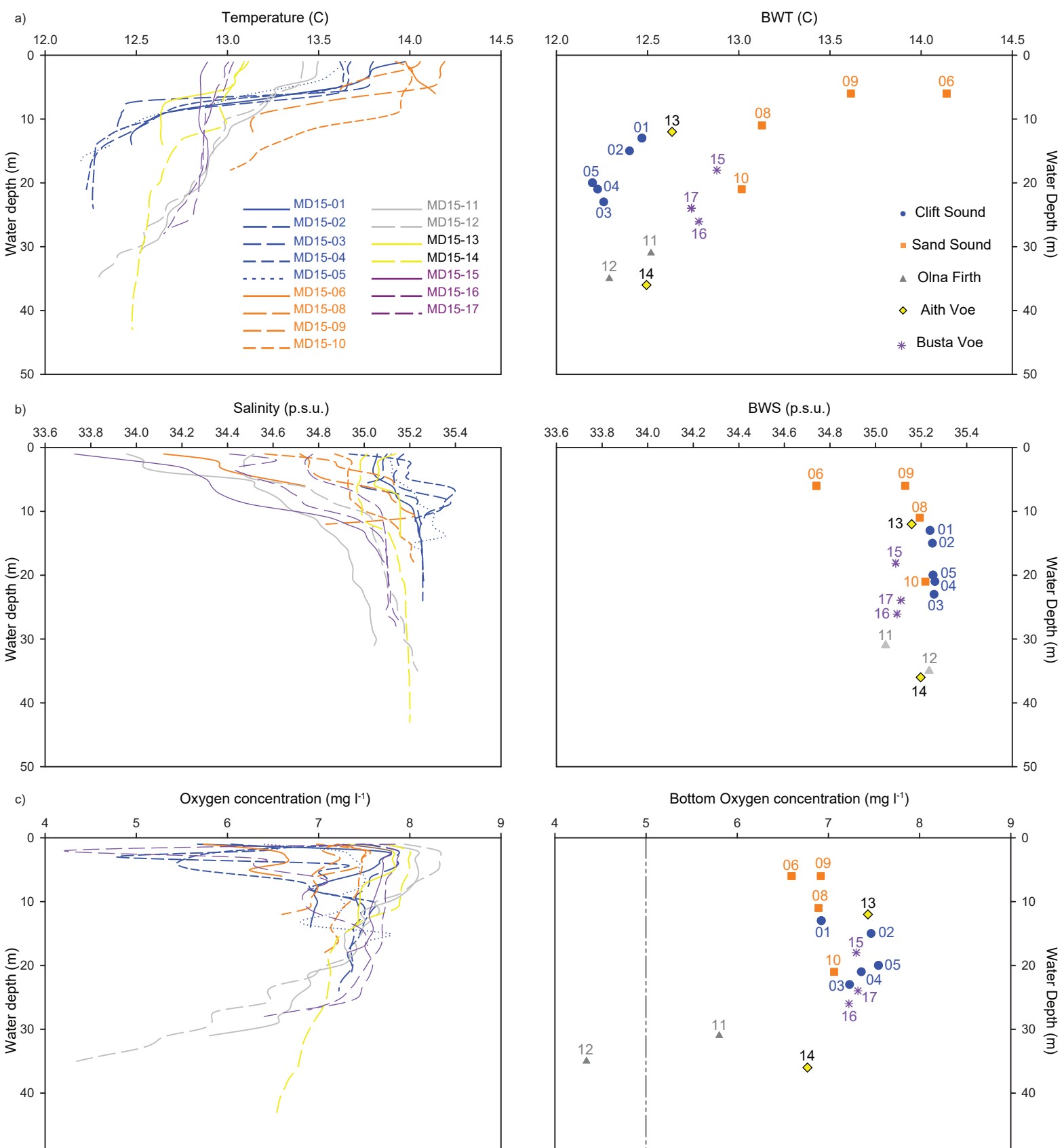

Fig. 2

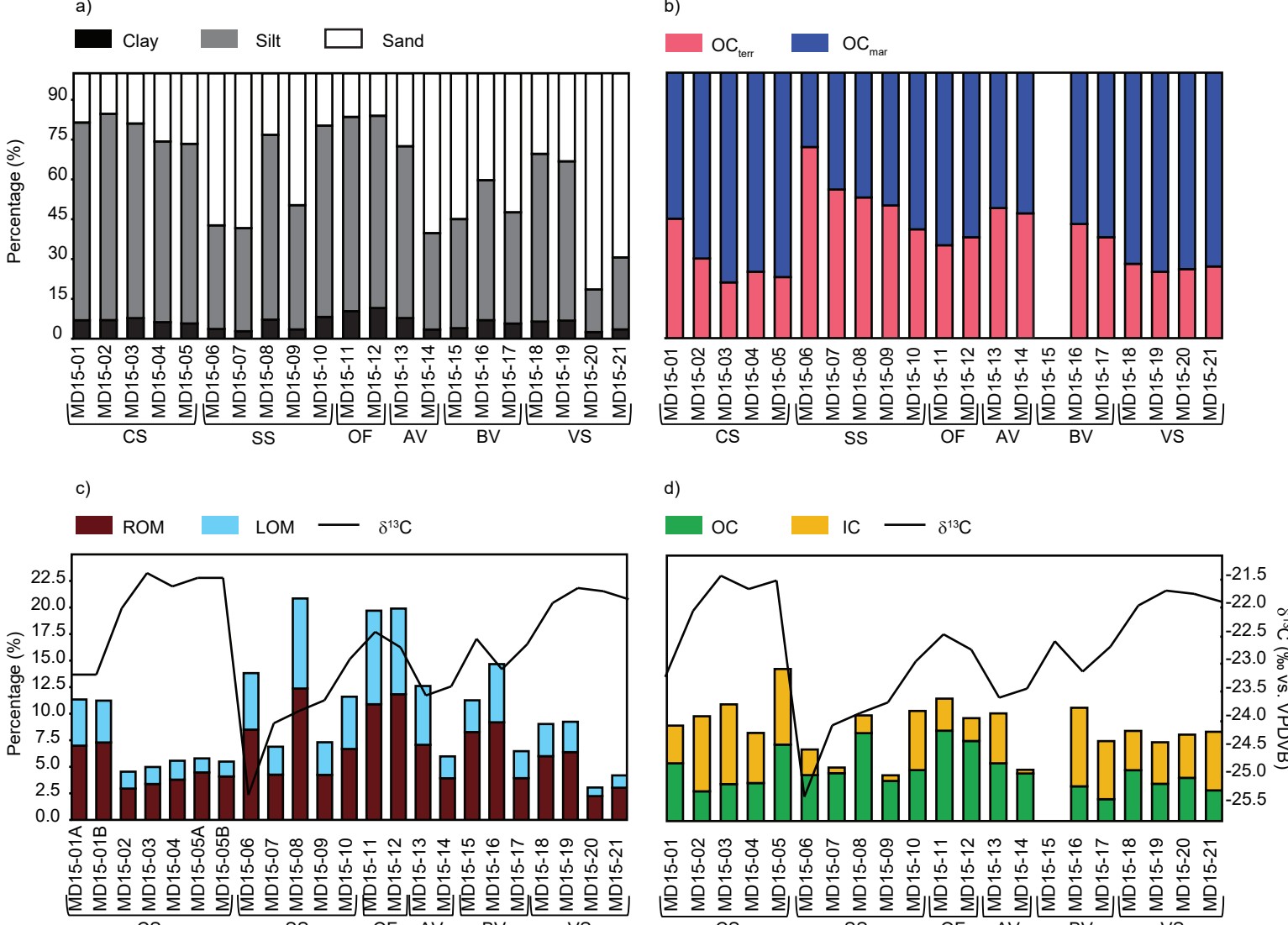

Fig. 3

a) Environmental parameters - Euclidean distance

b) Foraminifera relative abundance data - Bray-Curtis similarity

c) Canonical Component Analysis

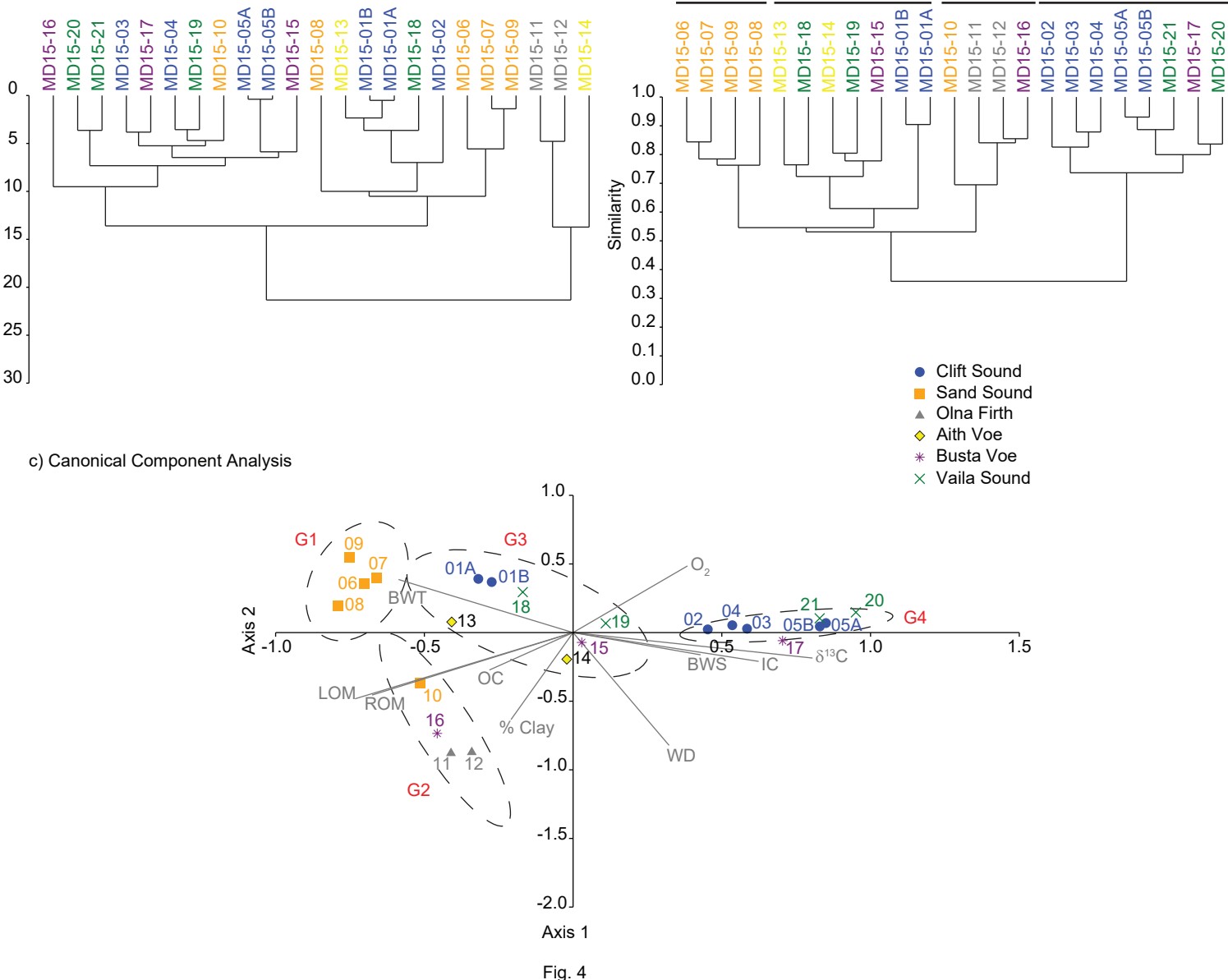

Fig. 4

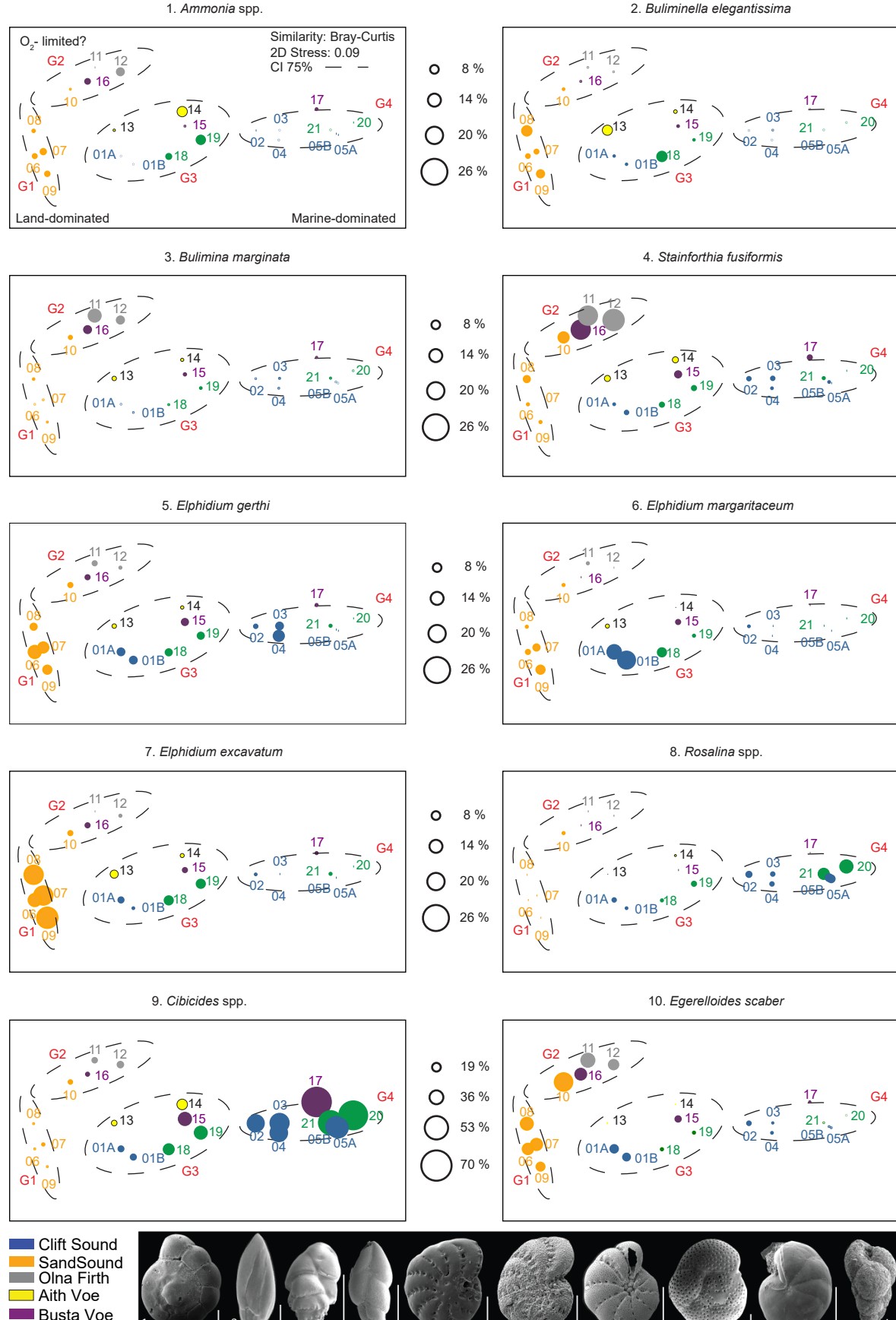

Fig. 5