# Peer review of "Organic carbon rich sediments: benthic foraminifera as bioindicators of depositional environments"

_Biogeosciences, 2019_

## Referee Comment (RC1) · Anonymous Referee #3 · 23 Jul 2019

This study attempts to reveal benthic foraminiferal responses to organic carbon in fjords system. Previous studies have revealed that benthic foraminifera are useful proxies for various environmental factors, such as oxygen content and food supply in many field. Each voe may have each sediment system because their bottom topography, surrounding environment, and river and current systems are different. These differences control the distributions of grain size, organic carbon, and benthic foraminifera. However, detailed environmental factors of each voe are not described in the manuscript. The authors discuss the "unrestricted" or "restricted" geomorphology, but figure 1 is quite small and detailed characteristics of each voe and sampling points are missing. There is no information about spatial distribution of grain size within each voe in Figure

3, so it is difficult to evaluate the distribution. Figures of spatial distributions of grain size, organic carbon, and benthic foraminiferal assemblage can assist in understanding the data interpretation. Grain size is another important parameter controlling benthic foraminiferal distribution in marine environments. However, the authors omit the grain size data from discussion of benthic foraminifera vs. environmental parameters for the reason that grain size distributions don't show obvious trend. Complex factors affect the benthic foraminiferal distributions in shallow marine environments, so I strongly recommend that the authors perform statistical analysis concerning the relationship between benthic foraminifera and environmental parameters (not only carbon but also grain size, water depth, BWS, and DO...).

Introduction Pg 2 Line 24-28: The authors mention the TROX model of Jorissen et al. (1995). This model is mainly applied to the deep sea setting because of its relatively stable environment. As mentioned above, shallow marine environment is affected by various environmental factors, so it is difficult to apply the simple relationship.

Pg 3 Line 7 (also for pg 8 Line 31, title of section 4.3, and pg 13 Line 4): The authors use the term "biogeography", but I think this study in not biogeographical study. "distribution" is more adequate than "biogeography".

Materials and methods Pg 3 Line 17: As mentioned above, the detailed descriptions of each voe should be included in the main body of manuscript.

Pg 3 Line 21-25: This part is methodology of foraminiferal analysis. I recommend to move this part to section 2.6.

Pg 3 Line 22-23: Why do you consider that the method of Schönfeld et al. (2012) may lead to underrepresentation of living foraminifera and the method is problematic? The authors explain that the reason for analyzing total (live + dead) assemblage is underrepresentation of living foraminifera in this part. However, the authors mention that the reason is "to provide a tool for the interpretation of fossil foraminiferal assemblages and their relationship with changes in OM and OC content in sediments over time" in page

5, line 23-24.

Pg 3 Line 26-27: The authors don't mention BWT. Please describe also about BWT (Fig. 2a).

Pg 4 Line 14: I think the detailed methodology of LOI analysis should be included in the main body of manuscript.

Pg 6 Line 3: "Fig. 4" should be "Fig. 5".

Results Pg 6 Line 26-27: TC data are not shown.

Pg 7 Line 5: The authors argue that high OC and low IC at the head of the voes, but I can't find this trend in figure 4a. Please also see comment below (comment to Pg 9 Line 2-6).

Pg 7 Line 7, 8: The authors use "Quinqueloculina seminulum" in supplementary table.

Pg 8 Line 3-4: Please add relative abundance after Cibicides spp. and E. scaber.

Discussion Pg 9 Line 2-6: The authors argue that a seaward gradient is evident, but I don't agree this argument. First of all, the authors analyze only two stations in Olna Firth and Aith Voe, and three stations in Busta Voe. So, you can't discuss the gradient in these voes. In addition, it seems that samples from Vaila Sound are not collected along the environmental gradient (Fig. 1). Moreover, it seems that there is no obvious seaward gradient in Clift Sound and Sand Sound.

Pg 9 Line 18-22: The authors argue that the high TC content at MD 15-05 is the effect of oil spill. If so, benthic foraminifera may change in response to the effect. Lei et al. (2015_Marine Pollution Bulletin) suggest that E. scaber is indicator of oils. Your foraminiferal data don't show the effect of oil or high TC.

References: Some articles are overlooked in references. Soil Survey of Scotland, 1981 Alve, 1994 Alve, 1995 Alve and Nagy, 1986 Qvale et al., 1984

Figure 1: Please add space between "Sand" and "Sound". As mentioned above, figure 1 is quite small and detailed characteristics of each voe and sampling points are missing.

Figure 2: Please add legends for each profile (i.e. please add "station number").

Figure 3: Please add legends for each plot (i.e. please add "station number"). It is difficult to identify the relationship between data and sampling point. I think figure 3b is not needed because the data are not discussed enough in the manuscript.

Figure 4: Distance of OCterr of Aith Voe (left one) don't match with OC and LOM. Third plot from the right (Clift Sound) is missing in figure of OCterr.

Figure 5: What are "Head" and "mouth" in the figure of Ammonia? Do you mean horizontal axis of MDS is gradient from head to mouth? Fig. 5-9 is Cibicidoides, but the authors use Cibicides in the text. The authors mention that "we grouped under the name E. excavatum both forma selseyense and forma clavata. . .", but "selseyensis" is used in figure 5-7. There is no indication of legend unit. Please add the unit (%?). There are no indication about four circled groups, so please add the name of each group near the dashed circles. It would be better to move SEM figures close to each MDS plot.

Supplementary Material: I can't find supplementary figures, so I can't evaluate supplementary figure 1 and 2. The authors use "Fig. 1a" in supplementary material, but figure 1 is single figure. The authors describe "the island of Vaila" and "the isle of Linga", but I can't identify the islands because there are no indication of these islands in figure 1.

Supplementary Table 1: The authors use Cibicidoides, but "Cibicides" is used in the text. Oolina mellow should be Oolina melo. Trochamina sp. should be Trochammina sp.. There are blanks only in the column of Reophax fusiformis. There is "0.0" in the gray cell of Reophax fusiformis. Please add picked number of specimens in table 1.

Best regards,

---

## Referee Comment (RC2) · Anonymous Referee #2 · 21 Aug 2019

Review of the ms "Organic carbon rich sediments: benthic foraminifera as bioindicators of depositional environments" by Elena Lo Giudice Cappelli et al.

The review is based on the version of the manuscript received in April 2019.

The aim of the present study is "To investigate the relationships between sedimentary OC in six west Shetland voes and the associated changes in benthic foraminiferal assemblages…" in order "…to: 1) Fingerprint the source (terrestrial vs. marine) and quality (refractory vs. labile) of organic matter and the form (organic vs. inorganic) of sedimentary carbon. 2) Establish benthic foraminiferal biogeography in Shetland's voes from recent surficial sediments. 3) Investigate the use of benthic foraminifera as bio-indicators of OC content in coastal sediments and their potential for palaeo-OC reconstruction purposes". This is a very topical theme, an interesting approach, and the manuscript should be of interest to the readers of Biogeosciences.

The manuscript is generally well organized and well written, all figures and tables are necessary, and adequate literature is cited. However, my concern is the weakly described quantitative relationship between the foraminiferal assemblages and the associated geochemical parameters. This relationship is supposed to serve as the baseline for using foraminifera as indicators for OC enrichment (see aims) and, hence, ought to be more clearly addressed. Methodological weaknesses (see examples below) which potentially affect the relationships/correlations should be identified and discussed.

Page 1, lines 13-14: "….. evaluate the use of modern benthic foraminifera as bio-indicators of carbon content in six voes (fjords) on the west coast of Shetland." I guess the authors do not mean any kind of carbon? Please specify. The same applies other places in the manuscript.

Page 1, lines 14-16: "Benthic foraminifera are sensitive….." Please make it clear to the reader if these statements are based on previous studies or results of the present study.

Page 3, line 20: "….sub-sampling the top layer of each grab,…" What was the thickness of the "top layer"? How do the authors know if the sampled top layers in the grabs were intact and had not lost some of the fines from the sediment-water interface, i.e. that the samples were comparable? Since no replicates were collected, how do the authors know how representative the OC and OM data were for each site?

Page 3, lines 22-23: "….foraminiferal counts are 'total' (live + dead) because the sampling technique may lead to underrepresentation of 'live' foraminifera." This needs some explanation.

Page 3, lines 26-27: "An earlier field survey of Shetland voes carried out in August 2009 measured bottom water temperature (BWT), salinity (BWS) and oxygen (O2) at the same locations as this study (Fig. 2)". If this implies that the present foraminiferal data collected in 2015 were only compared with

hydrographic data from 2009, it should be addressed in the discussion; particularly the statements postulating "low" or "poor" oxygen concentrations in Olna Firth, should be modified throughout the ms.

Page 5, lines 14-17: "Both size fractions were analysed. Depending on sample volume, we subdivided each sample into a number of splits using a standard splitter and, when possible, picked at least 300 specimens …". The samples were dry-sieved and dry-split? Please clarify. It is not clear why the samples were sieved into two size fractions? How did the authors ensure that the proportion between the two size fractions was the same in the counted splits as in the original sample? This is essential and needs to be explained.

Page 5, line 21: total assemblages (live + dead) were analysed. Please explain how you distinguished *in situ* tests from tests transported into the sites. This is particularly relevant in the more high energetic environments and deserves some comments.

Page 5, lines 23-26: "Ten taxa …." This belongs to results.

Page 6, lines 3-5: = results.

Page 6, lines 11-12: "…despite having very different geomorphologies (unrestricted vs. restricted) and circulation patterns (high vs. low energy) (Fig. 3)."  This belongs to discussion.

Page 7, line 3: "…at sites closed to land …" …… close to land

Page 7, section 3.1.4: Most of this belongs to discussion. How meaningful is the average stable isotope values of the different lochs? Would you not expect that the average values depend on how many samples are collected and analysed from different parts of the land-sea transect?

Page 8, line 7: "In Vaila Sound, an unrestricted geomorphology (Fig. 1), …"  It is not obvious, based on Fig. 1, that Vaila Sound has  an unrestricted geomorphology; please explain, and perhaps modify Fig. 1.

Page 8: Section 4.1 may be shortened, particularly since the data are not used in the further discussion.

Page 9, lines 24-32: These are results.

Page 10, lines 2 and 26: I cannot find the Supplementary Fig. 1 and Fig. 2.

Page 10, lines 4-5: "In general, foraminiferal assemblages do reflect the geomorphology of the six voes (restricted vs. unrestricted basins) and the seaward gradient in OM and OC distribution (Figs. 4 and 5)." The links between the foraminiferal assemblages and the distribution of OM and OC are neither easily seen from Figs 4 and 5, nor from the descriptions in the following sections. If the authors can show that the statement above actually holds, they should provide some clearer justifications.

Page 13, Conclusions, lines 28 and 32: The usefulness of benthic foraminifera as bio-indicators for OC is mentioned in the abstract, in the aims of the study, and in the conclusions but it is not addressed in the discussion. Hence, the importance of foraminifera as bio-indicators for OC in the present study should either be tuned down, or it should be thoroughly addressed in the discussion with concrete, quantitative, examples illustrating how they can be used.

Page 17, Fig. 2 caption. Please add that the CTD-data are from August 2009, whereas the sediment samples for the present study were collected in August 2015.

To summarize, this is a generally well written manuscript on a timely topic which should be of interest to the readers of Biogeosciences. The figures and tables are all needed and well-presented but some should be adjusted to show the postulated relationships between the benthic foraminiferal assemblages and the associated geochemical data. If possible, it would be helpful for the reader if Fig. 1 is modified so it indicates the difference between unrestricted and restricted geomorphologies of the voes.

I recommend publication of this manuscript following minor revision.

---

## Author Comment (AC1) · 10 Sep 2019

This study attempts to reveal benthic foraminiferal responses to organic carbon in fjords system. Previous studies have revealed that benthic foraminifera are useful proxies for various environmental factors, such as oxygen content and food supply in many field. Each voe may have each sediment system because their bottom topography, surrounding environment, and river and current systems are different. These differences control the distributions of grain size, organic carbon, and benthic foraminifera. However, detailed environmental factors of each voe are not described in the manuscript. The authors discuss the "unrestricted" or "restricted" geomorphology, but figure 1 is

quite small and detailed characteristics of each voe and sampling points are missing. There is no information about spatial distribution of grain size within each voe in Figure3, so it is difficult to evaluate the distribution. Figures of spatial distributions of grain size, organic carbon, and benthic foraminiferal assemblage can assist in understanding the data interpretation. Grain size is another important parameter controlling benthic foraminiferal distribution in marine environments. However, the authors omit the grain size data from discussion of benthic foraminifera vs. environmental parameters for the reason that grain size distributions don't show obvious trend. Complex factors affect the benthic foraminiferal distributions in shallow marine environments, so I strongly recommend that the authors perform statistical analysis concerning the relationship between benthic foraminifera and environmental parameters (not only carbon but also grain size, water depth, BWS, and DO: : :).

Following the referee's comments ,we have included (under the area of study section) a detailed description of available environmental parameters for each voe and modified Fig.1 to better represent the geomorphology of each sea loch (Page 3 lines 17 to page 4 line 18 and Fig. 1, additional info on soil types in Shetland are now illustrated in Suppl. Fig. 1). We modified Figs. 3 and 4 to better illustrate the spatial distribution of grain size, IC, OC and OM within each voe. We agree with the referee that grain size is an important parameter, generally co-varying with a number of other variables and therefore linked to foraminiferal distribution in marine environments; however, in our dataset we do not see this simple correlation, most likely because of the complexity of coastal systems compared to open marine settings. For example, Cibicides spp. occur in both fine and coarse sediments despite typically being a species that favours coarse substrates for its epilithic mode of attachment. To provide a more thorough representation of the relationship between benthic foraminiferal assemblage distribution and environmental parameters we performed on the full dataset (forams + 10 environmental parameters) canonical correspondence analysis including the % clay as one of the sedimentological parameters (Page 10 line 10 to page 11 line 3 and Figs. 3a and 4).

Introduction Pg 2 Line 24-28: The authors mention the TROX model of Jorissen et al. (1995). This model is mainly applied to the deep sea setting because of its relatively stable environment. As mentioned above, shallow marine environment is affected by various environmental factors, so it is difficult to apply the simple relationship.

Under the light of the referee's comment, we added that TROX models are typically used in deep-sea settings (Page 2 line 24) and that other environmental parameters like salinity and grain size distribution may be more variable and therefore affect the composition of benthic foraminiferal assemblages in shallow marine settings (par 4.3.5). This is further illustrated in Fig. 4.

Pg 3 Line 7 (also for pg 8 Line 31, title of section 4.3, and pg 13 Line 4): The authors use the term "biogeography", but I think this study in not biogeographical study. "distribution" is more adequate than "biogeography".

Changed into distribution pg.3 line 7 and recurrence.

Materials and methods Pg 3 Line 17: As mentioned above, the detailed descriptions of each voe should be included in the main body of manuscript.

Done. Page 3 line 17 to page 4 line 18.

Pg 3 Line 21-25: This part is methodology of foraminiferal analysis. I recommend to move this part to section 2.6.

Done. We revised paragraph 2.6 following the referee's recommendation and also the comments of Referee #2.

Pg 3 Line 22-23: Why do you consider that the method of Schönfeld et al. (2012) may lead to underrepresentation of living foraminifera and the method is problematic? The authors explain that the reason for analyzing total (live + dead) assemblage is under-representation of living foraminifera in this part. However, the authors mention that the reason is "to provide a tool for the interpretation of fossil foraminiferal assemblages and their relationship with changes in OM and OC content in sediments over time" in

page5, line 23-24.

We understand the referee's confusion and we have rephrased this part of the ms to clarify our points. We do not think that the method of Schoenfeld et al. (2012) may lead to underrepresentation of living foraminifera; what we meant is that scraping the top layer ($\sim$ 1 cm thick) of each grab with a domestic spoon may lead to a potential underestimation of living fauna if mixing of the top layer occurred when sampling. In this scenario, the number of living foraminifera at the sediment surface will be "diluted" due to the presence of dead/fossil specimens from deeper sediments. Having said this, a recent study by Rillo et al. (2019) reported that historical sediment samples collected in a way that could have caused disturbance of the sediment surface (sounding and dredge) are still representative of surface conditions and their foraminiferal assemblages can be used to reconstruct environmental changes reliably (Pg. 6 lines 18-29). The aim of this study is to provide a tool for the interpretation of fossil foraminiferal assemblages and their relationship with OC, therefore using total assemblages over living fauna is a more appropriate tool as total assemblages will better represent downcore fossil fauna found under comparable environmental conditions (revised par. 2.6).

Pg 3 Line 26-27: The authors don't mention BWT. Please describe also about BWT (Fig. 2a).

Done. Pg. 4 line 26.

Pg 4 Line 14: I think the detailed methodology of LOI analysis should be included in the main body of manuscript.

Done. Pg. 5 Lines 13-19.

Pg 6 Line 3: "Fig. 4" should be "Fig. 5".

Changed. Page 9 Line 16.

Results Pg 6 Line 26-27: TC data are not shown.

We changed Fig. 4 and TC data are now shown (Pg. 8 line 5 and Fig. 3d).

Pg 7 Line 5: The authors argue that high OC and low IC at the head of the voes, but I can't find this trend in figure 4a. Please also see comment below (comment to Pg 9 Line 2-6).

We agree with the referee and rephrased this part of the ms (Pg. 8 lines 10-11) and revised Fig. 4 (now Fig. 3). Due to the geomorphology and geometry of the voes, the pattern we observe in the distribution of OC and OM follows a proximity to land/freshwater input trend rather than a seaward gradient (Figs. 3d and 4).

Pg 7 Line 7, 8: The authors use "Quinqueloculina seminulum" in supplementary table.

Changed to seminula in Supplementary Table 1.

Pg 8 Line 3-4: Please add relative abundance after Cibicides spp. and E. scaber.

Done. Pg. 9 line 18.

Discussion Pg 9 Line 2-6: The authors argue that a seaward gradient is evident, but I don't agree this argument. First of all, the authors analyze only two stations in Olna Firth and Aith Voe, and three stations in Busta Voe. So, you can't discuss the gradient in these voes. In addition, it seems that samples from Vaila Sound are not collected along the environmental gradient (Fig. 1). Moreover, it seems that there is no obvious seaward gradient in Clift Sound and Sand Sound.

We agree with the referee and rephrased this part of the ms and revised Fig. 4. Due to the geomorphology and geometry of the voes (now better illustrated in Fig. 1), the pattern we observed in the distribution of OC and OM follows a proximity to land/freshwater input trend rather than a seaward gradient. We rephrased the discussions accordingly (par 4.2).

Pg 9 Line 18-22: The authors argue that the high TC content at MD 15-05 is the effect of oil spill. If so, benthic foraminifera may change in response to the effect. Lei et

al. (2015_Marine Pollution Bulletin) suggest that E. scaber is indicator of oils. Your foraminiferal data don't show the effect of oil or high TC.

We agree with the referee that foram assemblages should reflect the oil spill; it was just a speculation given the surprisingly high levels of OC at this location – not reflected in the Cibicides dominated assemblages. We revised this speculation and left an open question regarding what could be driving the high TC content at station MD15-05 (Pg. 11 lines 21-28).

References: Some articles are overlooked in references. Soil Survey of Scotland, 1981 Alve, 1994 Alve, 1995 Alve and Nagy, 1986 Qvale et al., 1984

Fixed.

Figure 1: Please add space between "Sand" and "Sound". As mentioned above, figure 1 is quite small and detailed characteristics of each voe and sampling points are missing.

Changed. Fig. 1.

Figure 2: Please add legends for each profile (i.e. please add "station number").

Done. Fig.2.

Figure 3: Please add legends for each plot (i.e. please add "station number"). It is difficult to identify the relationship between data and sampling point. I think figure 3b is not needed because the data are not discussed enough in the manuscript.

We substantially revised Fig. 3 to include the referee's suggestions, also based on previous comments.

Figure 4: Distance of OCterr of Aith Voe (left one) don't match with OC and LOM. Third plot from the right (Clift Sound) is missing in figure of OCterr. We substantially revised Fig. 4 to include the referee's suggestions, also based on previous comments.

This info in now in the revised Fig. 3.

Figure 5: What are "Head" and "mouth" in the figure of Ammonia? Do you mean horizontal axis of MDS is gradient from head to mouth? Fig. 5-9 is Cibicidoides, but the authors use Cibicides in the text. The authors mention that "we grouped under the name E. excavatum both forma selseyense and forma clavata: : :", but "selseyensis" is used in figure 5-7. There is no indication of legend unit. Please add the unit (%?). There are no indication about four circled groups, so please add the name of each group near the dashed circles. It would be better to move SEM figures close to each MDS plot.

We revised Fig. 5 to include the above suggestions; however, it was not possible to include the SEM figure close to each MDS plot due to legibility issues. Foram nomenclature is now consistent throughout the revised manuscript.

Supplementary Material: I can't find supplementary figures, so I can't evaluate supplementary figure 1 and 2.

We apologise for the missing supplementary material. There must have been a problem when we uploaded those files and we were not aware they were missing. We have now included both supplementary figures.

The authors use "Fig. 1a" in supplementary material, but figure 1 is single figure. The authors describe "the island of Vaila" and "the isle of Linga", but I can't identify the islands because there are no indication of these islands in figure 1.

We revised Fig. 1 to better illustrate the geometry and geomorphology of each voe.

Supplementary Table 1: The authors use Cibicidoides, but "Cibicides" is used in the text. Oolina mellow should be Oolina melo. Trochamina sp. should be Trochammina sp.. There are blanks only in the column of Reophax fusiformis. There is "0.0" in the gray cell of Reophax fusiformis. Please add picked number of specimens in table 1.

We revised Supplementary Table 1 following the referee's remarks. Upon acceptance

of this ms, the full dataset will be made available on PANGEA.

Please also note the supplement to this comment:
https://www.biogeosciences-discuss.net/bg-2019-125/bg-2019-125-AC1-
supplement.pdf

[Figure]

Fig. 1

**Fig. 1.**

[Figure]

Fig. 2

**Fig. 2.**

[Figure]

Fig. 3

**Fig. 3.**

[Figure]

Fig. 4.

[Figure]

Fig. 5.

**Fig. 5.**

[Figure]

**Fig. 6.**

---

## Author Comment (AC2) · 10 Sep 2019

Review of the ms "Organic carbon rich sediments: benthic foraminifera as bioindicators of depositional environments" by Elena Lo Giudice Cappelli et al.

The review is based on the version of the manuscript received in April 2019.

The aim of the present study is "To investigate the relationships between sedimentary OC in six west Shetland voes and the associated changes in benthic foraminiferal assemblages..." in order "...to: 1) Fingerprint the source (terrestrial vs. marine) and quality (refractory vs. labile) of organic matter and the form (organic vs. inorganic)

of sedimentary carbon. 2) Establish benthic foraminiferal biogeography in Shetland's voes from recent surficial sediments. 3) Investigate the use of benthic foraminifera as bioindicators of OC content in coastal sediments and their potential for palaeoâÅROC reconstruction purposes". This is a very topical theme, an interesting approach, and the manuscript should be of interest to the readers of Biogeosciences. The manuscript is generally well organized and well written, all figures and tables are necessary, and adequate literature is cited. However, my concern is the weakly described quantitative relationship between the foraminiferal assemblages and the associated geochemical parameters. This relationship is supposed to serve as the baseline for using foraminifera as indicators for OC enrichment (see aims) and, hence, ought to be more clearly addressed. Methodological weaknesses (see examples below) which potentially affect the relationships/correlations should be identified and discussed. Following the referee's suggestions, we strengthen the discussion regarding the use of benthic foraminifera as indicators of OC content in marine sediments with emphasis on the relationship between forams and geochemical parameters (par 3.2.1 and par. 4.3). We additionally revised the methods and discussed potential weaknesses that could affect the strength of the relationship between forams and OC.

Page 1, lines 13-14: "..... evaluate the use of modern benthic foraminifera as bio-indicators of carbon content in six voes (fjords) on the west coast of Shetland." I guess the authors do not mean any kind of carbon? Please specify. The same applies other places in the manuscript. We added organic before carbon (Pg. 1 line 14 and following instances).

Page 1, lines 14-16: "Benthic foraminifera are sensitive....." Please make it clear to the reader if these statements are based on previous studies or results of the present study. We rephrased this sentence to make clear that the statement is based on previous studies (Pg. 1 lines 14-16).

Page 3, line 20: "....sub-sampling the top layer of each grab,..." What was the thickness of the "top layer"? How do the authors know if the sampled top layers in the

BGD
grabs were intact and had not lost some of the fines from the sedimentâÅŘwater interface, i.e. that the samples were comparable? Since no replicates were collected, how do the authors know how representative the OC and OM data were for each site? The thickness of the top layer is  $\sim$  1 cm (Pg. 4 line 21). We did not have a way of controlling for loss of fine material at the sediment-water interface when collecting grab samples; however, our results compare well with the grain size distribution found in other Scottish sea lochs (Pg. 10 lines 28-31) suggesting that if there were a loss of fines, it was most likely negligible. We did not have replicate samples for OC measurements, but we did have two pairs of replicate samples for OM which resulted in a mean relative error of  $\pm$  0.07 % for LOM,  $\pm$  0.06 % for ROM and  $\pm$  0.03 % for TOM (Pg. 5 lines 17-19), pointing to a very good reproducibility of data and representation of local conditions.

Page 3, lines 22-23: "....foraminiferal counts are 'total' (live + dead) because the sampling technique may lead to underrepresentation of 'live' foraminifera." This needs some explanation. Following this suggestion and comments of Referee #3, we revised this part of the ms to improve clarity (Par. 2.6).

Page 3, lines 26âÅŘ27: "An earlier field survey of Shetland voes carried out in August 2009 measured bottom water temperature (BWT), salinity (BWS) and oxygen (O2) at the same locations as this study (Fig. 2)". If this implies that the present foraminiferal data collected in 2015 were only compared with 2 hydrographic data from 2009, it should be addressed in the discussion; particularly the statements postulating "low" or "poor" oxygen concentrations in Olna Firth, should be modified throughout the ms. We agree with the referee in that BWT, BWS and O2 in 2009 may be different from those of 2015 and we have now discussed this possibility in our ms (Pg. 9 lines 25-27; Pg. 12 lines 27-29; Pg. 13 lines 22-24).

Page 5, lines 14-17: "Both size fractions were analysed. Depending on sample volume, we subdivided each sample into a number of splits using a standard splitter and, when possible, picked at least 300 specimens ...". The samples were dry-sieved and dry-split? Please clarify. It is not clear why the samples were sieved into two
size fractions? How did the authors ensure that the proportion between the two size fractions was the same in the counted splits as in the original sample? This is essential and needs to be explained. Samples were dry-sieved and dry-split. We added in the methods that samples were split in the two size fractions prior to this study and that we analysed both size fractions to make sure that no environmental information was lost due to changes in biodiversity and/or in the composition of benthic foraminiferal assemblage between the two size fractions (Pg. 6 line 30 to Pg. 7 line 9).

Page 5, line 21: total assemblages (live + dead) were analysed. Please explain how you distinguished in situ tests from tests transported into the sites. This is particularly relevant in the more high-energetic environments and deserves some comments. We had no means to systematically distinguish in situ tests from advected ones when counting unstained specimens. What we did was to compare our assemblages with published data from other Scottish locations and fjords to detect possible inconsistences that may suggest advection of material from other locations, especially in the more energetic environments. Our data overall compare well with other studies, suggesting that advection of tests from other locations is negligible in our sample set. We included this information in Pg. 15 lines 19-21.

Page 5, lines 23-26: "Ten taxa ...." This belongs to results. Moved. Pg. 9 Lines 12-15.

Page 6, lines 3-5: = results. Moved. Pg. 9 Lines 15-16.

Page 6, lines 11-12: "...despite having very different geomorphologies (unrestricted vs. restricted) and circulation patterns (high vs. low energy) (Fig. 3)." This belongs to discussion. Removed.

Page 7, line 3: "...at sites closed to land ..." ..... close to land Changed. Pg. 8 Line 13.

Page 7, section 3.1.4: Most of this belongs to discussion. How meaningful is the aver-
age stable isotope values of the different lochs? Would you not expect that the average values depend on how many samples are collected and analysed from different parts of the land-sea transect? Moved to discussion and included range of isotope values and/or number of measurements when possible (Pg. 14 lines 17-20).

Page 8, line 7: "In Vaila Sound, an unrestricted geomorphology (Fig. 1), ..." It is not obvious, based on Fig. 1, that Vaila Sound has an unrestricted geomorphology; please explain, and perhaps modify Fig. 1. We modified Fig. 1 and this part of the discussion.

Page 8: Section 4.1 may be shortened, particularly since the data are not used in the further discussion. Following the comments of Referee #3 we now included % Clay in our discussion, revised Figs. 3 and 4 accordingly, and moved to the supplement Fig. 3b (now supplementary fig. 2).

Page 9, lines 24-32: These are results. Moved into a new paragraph in the results (par 3.2.1).

Page 10, lines 2 and 26: I cannot find the Supplementary Fig. 1 and Fig. 2. We apologise for the missing of supplementary material. There must have been a problem when we uploaded those files as we were not aware they were missing. We have now included both supplementary figures.

Page 10, lines 4âÅŘ5: "In general, foraminiferal assemblages do reflect the geomorphology of the six voes (restricted vs. unrestricted basins) and the seaward gradient in OM and OC distribution (Figs. 4 and 5)." The links between the foraminiferal assemblages and the distribution of OM and OC are neither easily seen from Figs 4 and 5, nor from the descriptions in the following sections. If the authors can show that the statement above actually holds, they should provide some clearer justifications. Under the light of the referee's comment, we added canonical correspondence analysis of our dataset (foraminifera relative abundance + 10 env parameters: WD, BWT, BWS, O2, IC, OC, LOM, ROM, ïĄd'13C, % Clay) to better constrain the relationship between environmental parameters and benthic foraminifera assemblage distribution and better BGD
illustrate the links between forams and OC (par. 3.2.1 and par. 4.3 and Fig. 4)

Page 13, Conclusions, lines 28 and 32: The usefulness of benthic foraminifera as bio-indicators for OC is mentioned in the abstract, in the aims of the study, and in the conclusions but it is not addressed in the discussion. Hence, the importance of foraminifera as bio-indicators for OC in the present study should either be tuned down, or it should be thoroughly addressed in the discussion with concrete, quantitative, examples illustrating how they can be used. Following the referee's comment, we expanded the discussion about the relationship between forams and OC to better illustrate how benthic foraminiferal assemblages can be used as indicators of past changes in OC deposition and accumulation in marine sediments.

Page 17, Fig. 2 caption. Please add that the CTD-data are from August 2009, whereas the sediment samples for the present study were collected in August 2015. Done. Pg. 20 Line 11.

To summarize, this is a generally well written manuscript on a timely topic which should be of interest to the readers of Biogeosciences. The figures and tables are all needed and well-presented but some should be adjusted to show the postulated relationships between the benthic foraminiferal assemblages and the associated geochemical data. If possible, it would be helpful for the reader if Fig. 1 is modified so it indicates the difference between unrestricted and restricted geomorphologies of the voes. We thank the referee for constructive comments and this commendation. We have modified Fig. 1 to better illustrate the geomorphology of each voe and extended the discussion regarding the links between foram assemblages and environmental parameters. Additionally, Suppl. Fig. 1 illustrates type of soils in Shetland and % TOC in soils.

Please also note the supplement to this comment: https://www.biogeosciences-discuss.net/bg-2019-125/bg-2019-125-AC2supplement.pdf BGD
Fig. 1.

Fig. 2.

**Supplement:**

[Figure]

Supplementary Fig. 1

[Figure]

Supplementary Fig. 3

**Supplementary Figure 1. Shetland soil maps.** Map on the left shows soil types in Shetland, map on the right illustrates TOC content in Shetland's soils.

**Supplementary Figure 2. SEM micrographs of benthic foraminifera in six voes on Shetland's west coast; scale bar = 100 μm; identification after Austin (1991) and references therein. 1** *Ammonia batavus*. **2** *Ammonia* sp. **3** *Ammoscalaria runiana*. **4** *Bolivina pseudoplicata*. **5** *Bolivina skagerrakensis*. **6** *Bolivinellina pseudopunctata*. **7** *Buccella frigida*. **8** *Buccella tenerrima*. **9** *Bulimina elongata*. **10** *Bulimina marginata*. **11** *Buliminella elegantissima*. **12** *Cassidulina laevigata*. **13** *Cassidulina obtusa*. **14** *Cibicides lobatulus*. **15** *C. lobatulus*. deformed. **16** *Connemarella rudis*. **17** *Discanomalina* sp. **18** *Eggerelloides scaber*. **19** *Elphidium aculeatum*. **20** *Elphidium albiumbelicatum*. **21** *Elphidium crispum*. **22** *Elphidium excavatum* (subsp clavatum). **23** *Elphidium excavatum* (subsp selseyense). **24** *Elphidium gerthi*. **25** *Elphidium incertum*. **26** *Elphidium margaritaceum*. **27** *Elphidium* sp. **28** *Elphidium williamsoni*. **29** *Epistominella vitrea* (*Eilohedra vitrea*). **30** *Oolina hexagona* (*Favulina hexagona*). **31** *Oolina melo* (*Favulina melo*). **32** *Fissurina* sp. **33** *Haplophragmoides* sp. **34** *Haynesina germanica*. **35** *Hyalina balthica*. **36** *Glandulina* sp. **37** *Lagena setigera*. **38** *Lagena* sp. **39** *Lagena striata*. **40** *Miliolinella subrotunda*. **41** *Nonion* sp. **42** *Nonion* sp. **43** *Nonionoides turgidus*. **44** *Oolina williamsoni* (*Homalohedra williamsoni*). **45** *Patellina corrugata*. **46** *Planorbulina distoma*. **47** *Quinqueloculina bicornis* (*Adelosina bicornis*). **48** *Quinqueloculina seminula*. **49** *Reophax fusiformis*. **50** *Quinqueloculina* sp. **51** *Rosalina anomala*. **52** *Rosalina* sp. **53** *Spirillina vivipara*. **54** *Spiroloculina rotunda*. **55** *Spiroplectinella wrightii*. **56** *Stainforthia fusiformis*. **57** *Subanomalina pauperata*. **58** *Textularia earlandi*. **59** *Trochammina* sp. **60** Idetermined. **61** Indetermined. **62** Indetermined. **63** Indetermined. **64** Etched *Ammonia* sp.

**Supplementary Figure 3.** Trends between particle size and organic matter and carbon content.